# Rapid activation of distinct members of multigene families in *Plasmodium spp*

Radoslaw Igor Omelianczyk [1,6], Han Ping Loh [1,6], Marvin Chew [1,2], Regina Hoo[1],
Sebastian Baumgarten [3], Laurent Renia [4], Jianzhu Chen [2,5] & Peter R. Preiser[1,2✉]

The genomes of *Plasmodium spp.* encode a number of different multigene families that are thought to play a critical role for survival. However, with the exception of the *P. falciparum var* genes, very little is known about the biological roles of any of the other multigene families. Using the recently developed Selection Linked Integration method, we have been able to activate the expression of a single member of a multigene family of our choice in *Plasmodium spp.* from its endogenous promoter. We demonstrate the usefulness of this approach by activating the expression of a unique *var, rifin* and *stevor* in *P. falciparum* as well as *yir* in *P. yoelii*. Characterization of the selected parasites reveals differences between the different families in terms of mutual exclusive control, co-regulation, and host adaptation. Our results further support the application of the approach for the study of multigene families in *Plasmodium* and other organisms.

[1] School of Biological Sciences, Nanyang Technological University, Singapore, Singapore. [2] Singapore-MIT Alliance for Research and Technology, Infectious Disease Interdisciplinary Research Group, Singapore, Singapore. [3] Biology of Host–Parasite Interactions Unit, Department of Parasites and Insect Vectors, Institut Pasteur, CNRS ERL 9195, INSERM Unit U1201, Paris, France. [4] Singapore Immunology Network (SIgN), A*STAR, Biopolis, Singapore, Singapore. [5] Koch Institute for Integrative Cancer Research and Department of Biology, Massachusetts Institute of Technology, Cambridge, MA, USA. [6] These authors contributed equally: Radoslaw Igor Omelianczyk, Han Ping Loh. ✉email: prpreiser@ntu.edu.sg

The malaria parasite exports a large number of proteins into the host red blood cell to modify it for its own survival[1–3]. Among these proteins are variant surface antigens (VSAs) which are displayed on the surface of the infected cell. One of the major hurdles in the study of these proteins is their variable nature as they are encoded by large multigene families[4,5]. VSAs can be broadly grouped into the high molecular weight and *P. falciparum* specific PfEMP1 (coded by *var* genes)[6] and the small variant surface antigens (sVSAs) which are usually about 30–40 kDa in size and found in all *Plasmodium spp*[5,7–10]. Within this group are the *P. falciparum rifin* and *stevor* families (coding for RIFIN and STEVOR proteins, respectively) as well as the *P. yoelii yir* (coding for YIR proteins) multigene family.

In *Plasmodium*, multigene families are commonly found near the telomeric ends of each chromosome and are organized as heterochromatin in distinct clusters at the periphery of the nucleus[11–14]. Genomic variation among *var* genes is generated through ectopic recombination during mitosis[14–16]. Even though it has been shown that multigene families in other organisms such as the VSGs from *Trypanosoma brucei*, serotypes of *Borrelia hermsii* or mating types in *Saccharomyces cerevisiae* show high levels of regulation[17–19], only *var* genes seem to show the same level of mutual exclusive control. Under normal circumstances only one *var* gene is expressed per cell, though this can change under certain circumstance with a recent study describing parasites expressing two[20,21]. The exact mechanism on how mono-allelic expression is controlled is not clearly understood, but it involves different layers including transcription factors, untranslated ORFs, 5′ upstream regions, histone modifications and nuclear localization[22–30]. Small VSAs show a less strict level of control. The current consensus is that expression of *rifin* is not mutually exclusive[31–33] with one to a few *rifin* being expressed per cell[34,35]. However, phenotypic selection resulted in the selection of a single *rifin*[35], suggesting some mutual exclusive control after all. *Rifins* are regulated through histone modifications and nuclear localization[36–38]. Expression of *stevors* is restricted to 1–3 genes per cell[39,40] and no information on epigenetic control mechanisms is available for this family. To date there is no evidence that supports a potential co-regulation between different *P. falciparum* multigene families[41].

Analogous to the epigenetic control, most of our understanding of the function of VSAs is based on numerous studies carried out on a limited number of different PfEMP1 proteins of *P. falciparum*. In addition to a predicted role in host immune evasion[42,43], it has been shown that certain members of the *var* multigene family bind to receptors on the surface of human red blood cells or endothelial cells. Among those are chondroitin sulfate (CSA), CD36, thrombospondin, intercellular adhesion molecule1(ICAM1) and complement receptor 1 (CR1)[44–46]. Only recently did new data start to shed light on the important biological roles of the STEVOR and RIFIN multigene families with both binding to uninfected red blood cells through interaction with glycophorin C or blood group A antigen, respectively[35,47]. Furthermore, certain RIFINs are interacting with leucocyte immunoglobulin-like receptor B1 (LILRB1) or leucocyte-associated immunoglobulin-like receptor 1 thereby actively modulating the host immune response[48]. In *P. vivax* VIRs have been suggested to be involved in binding to endothelial cells through interaction with ICAM1[49] while in the rodent *Plasmodium* species, *P. chabaudi* CIRs have been shown to interact with rodent red blood cells[50].

Despite their clear importance for the parasite only a very small number of VSA members have been studied to date, therefore greatly limiting our understanding of the interplay between VSA expression and pathology and disease outcome. This is partially due to the experimental difficulty faced when working with VSAs.

Due to the large copy number of VSAs encoding genes along with the high sequence similarity, development of specific genetic approaches or unique reagents to study these multigene families effectively faces particular problems. In *P. falciparum* this is further complicated by the fact that many multigene families lose their expression in vitro after long term culture[51,52] and the lack of any suitable functional screen. Currently, the study of PfEMP1 proteins relies on time consuming methods such as panning, which is limited to genes which can be selected for by these methods[53,54]. Episomal expression is especially difficult due to the size of full-length PfEMP1 proteins. Furthermore, constitutive promoters used in episomal plasmids might cause proteins to be expressed at the wrong time, causing misleading results and constructs using the promoter regions of multigene members coupled to drug resistance markers can alter the epigenetic landscape[25,31,55]. In vivo, *P. falciparum* as well as rodent parasites switch VSA expression rapidly from the parental cloned parasite[33,56,57] making it particularly difficult to study the function of multigene families in parasites that currently can only be maintained in vivo including the human parasite *P. vivax*, and rodent malaria models for *P.yoelii*, *P. chabaudi* and *P. berghei*[58–61].

In this study, we have adapted an experimental approach to overcome some of these critical challenges, thereby opening the potential to initiate a comprehensive and systematic study of VSAs. We demonstrated that with this technique we can express a single variant of the *var* or *stevor* gene families. Furthermore, we successfully extended the use of this technique for the functional characterization of other multigene families in both in vivo and in vitro settings.

## Results

**Activating single multigene family members in *P. falciparum*.** Activation of a specific member of a *Plasmodium* multigene family from its endogenous promoter has so far only been achieved in the case of the *var* family, where the knowledge of the distinct adhesive properties of some PfEMP1 members was used as the basis of functional enrichment[44–46]. However, this approach only can be applied for those PfEMP1 members whose adhesive properties are known. To overcome this limitation we have adapted the recently published Selection Linked Integration (SLI) method[62] to activate distinct members of each of the *var*, *rifin* and *stevor* multigene families of *P. falciparum*. This method is based on tagging endogenous genes with a neomycin resistance cassette, resulting in co-expression of the tagged endogenous gene and introduced selectable marker from a single transcript separated by a ribosomal skip peptide (Fig. 1). This approach ensures that resistant parasites are only obtained if the gene targeted is transcriptionally active. To validate this method, we generated parasites that tagged either a unique *var* (Pf3D7_0421300) or *rifin* (Pf3D7_1254800) with GFP and a unique *stevor* (Pf3D7_1149900) with 3HA. To the best of our knowledge only Pf3D7_1254800 has been studied so far[48], with no information about Pf3D7_0421300 or Pf3D7_1149900 available.

After selection of resistant parasites, integration was confirmed by PCR and whole genome sequencing (Supplementary Figs. 1 and 5a). Correct expression and complete ribosomal cleavage of the fusion protein was assessed using Western blot analysis (Fig. 2, Supplementary Fig. 5b). Percoll-enriched late stage parasites were separated into host and parasite fraction using saponin lysis with proteins exported into the host cell cytoplasm (HCC) expected to be detected in both fractions. Aldolase, a cytosolic parasite protein, served as a fractionation control[63]. All three cell lines showed bands of the expected size of the fusion protein with an ~42 kDa band being detected for STEVOR, an ~280 kDa for PfEMP1 and an ~66 kDa for RIFIN. In the PfEMP1

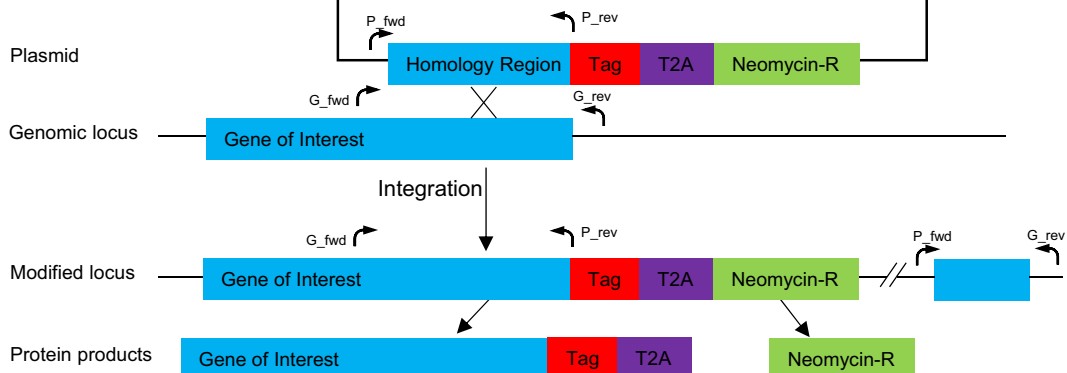

**Fig. 1 Schematic of Selection Linked Integration method.** The plasmid includes a ~700 bp long single recombination arm of the 3′ end of the gene of interest (without stop codon), followed by a tag of choice, a T2A skip peptide and a neomycin resistance cassette. During translation, the growing amino acid chain breaks at the end of the T2A peptide, resulting in two independent proteins from a single mRNA molecule. Arrows indicate location of primers used for integration validation.

knock-in cell line an additional 100 kDa band was detected in the exported fraction likely representing a proteolytic cleavage occurring during export into the HCC. In addition, while the STEVOR and PfEMP1 proteins appear in both the parasite as well as HCC fractions, RIFIN is only detectable within the parasite suggesting that this protein is not exported.

To further investigate the intracellular localization of the tagged proteins, immune fluorescence assays (IFA) of cells fixed with either methanol (trophozoite stage) or 4% paraformaldehyde (merozoites) was carried out using antibodies against their respective tags (Fig. 2). Co-staining with either EXP2, a parasitophorous membrane (PVM) resident protein that demarcates the boundaries of the parasite[64] and sEMP1, a Maurer's Cleft resident protein[65], was carried out to establish export and location within the HCC. The tagged STEVOR was detected in punctate dots beyond the PVM at the trophozoite stage (Fig. 2a), that co-stained with sEMP1. No tagged STEVOR could be detected in merozoites that where co-stained with Gap45, a member of the glideosome[66] (Fig. 2a). The tagged PfEMP1-showed a pattern similar to STEVOR with most of the protein co-localizing with sEMP1 in Maurer's Cleft while some staining is also detected inside the parasite (Fig. 2b). Some diffuse staining of PfEMP1 can also be observed in the merozoite and could represent the residual amount of PfEMP1 that was not exported (Fig. 2b). Mild tryptic digest of Percoll-enriched late stage parasites confirmed presentation of the tagged PfEMP1 on the surface of the infected red blood cell (Supplementary Figs. 3a and 5c). In contrast, the tagged RIFIN remained entirely within the parasite (Fig. 2c) and showed a distinct staining at the apical end of the merozoite where it partially co-localized with Gap45 (Fig. 2c). The observed location for PfEMP1 and STEVOR is in line with previous data showing export of the proteins to Maurer's cleft and beyond[67] while studies of RIFIN have indicated that a subset of them are expressed within the parasite and located at the apical end of the merozoite[68].

**Changes in multigene expression upon forced expression.** Monoallelic expression of *var* has been clearly demonstrated; however, whether this also applies to other *Plasmodium* multigene families has not yet been fully established. The selection linked activation used here not only allows activation of a single member of a multigene family from its endogenous promoter but is also ideally suited to evaluate potential co-regulation of expression both within the same gene family but also other gene families within the same cell. To confirm that the selection linked activation indeed results in monoallelic expression of *var* as well

as to gain insights about possible co-regulation of *stevor* as well as *rif*, we carried out microarray analysis on all three knock-in cell lines and parental 3D7 at three different timepoints (18, 24 and 30 hpi), covering the peak expression of all three multigene families studied (Fig. 3).

The activation of *var*, *rif* and *stevor* led to pronounced changes in overall multigene expression pattern. Compared to parental 3D7, activation of *stevor* (Pf3D7_1149900) resulted in strong transcription of the targeted gene while all other *stevor* were strongly downregulated (Fig. 3b). In contrast, activation of a specific *rif* or *var* resulted in either several *stevors* or in the case of *var* activation a single *stevor* being highly expressed. However, in all these parasites the overall expression levels of *stevor* were lower than in the *stevor*-activated cell line.

Activation of *var* (Pf3D7_0421300) resulted in elevated transcription of the targeted gene as well as a second *var*, Pf3D7_0421100 (Fig. 3a). Both genes are located in a head to tail manner on the same strand of chromosome 4, separated by 3.3 kbps of DNA, suggesting that the epigenetic mechanisms that govern active *var* expression open up the chromatin structure and enable expression of the neighbouring *var* as well (Supplementary Fig. 2a). In the *stevor*-activated cell line it is noteworthy that while there are still a range of *var* being transcribed there is a striking activation of a single *var* (Pf3D7_0617400), suggesting some possible co-regulation between *stevor* and *var*. Activation of a specific *rif* on the other hand appears to have no impact on the transcriptional status of *var* (Fig. 3a).

In striking contrast, activation of the *rif* (Pf3D7_1254800) did not greatly affect overall *rifin* expression (Fig. 3c), with several genes being highly transcribed. In fact, the targeted gene is not among the highest expressed *rif* suggesting the *rif* multigene family is not under mutual exclusive expression control. It is though interesting to note that the activation of *var* (Pf3D7_0421300) led to the concurrent activation of the adjacent *rif* (Pf3D7_0421500). The genes are oriented in a head to tail manner and separated by about 5.1 kbps which include the non-coding RNA RUF6. Another *rif* (Pf3D7_0421200), which is located 800 bps downstream of the targeted *var* in a tail to tail fashion was apparently not affected (an overview of the genomic locus surrounding Pf3D7_0421300 can be found in Supplementary Fig. 2a). A similar activation of an adjacent *rif* (Pf3D7_1149800) was also seen in the case of the *stevor*-activated parasite line (Fig. 3c).

In order to verify the microarray data and show that the expression remains stable we performed qRT-PCR on 15 different multigenes (5 *var*, 5 *rifin*, 5 *stevor*) 6 weeks after the microarray

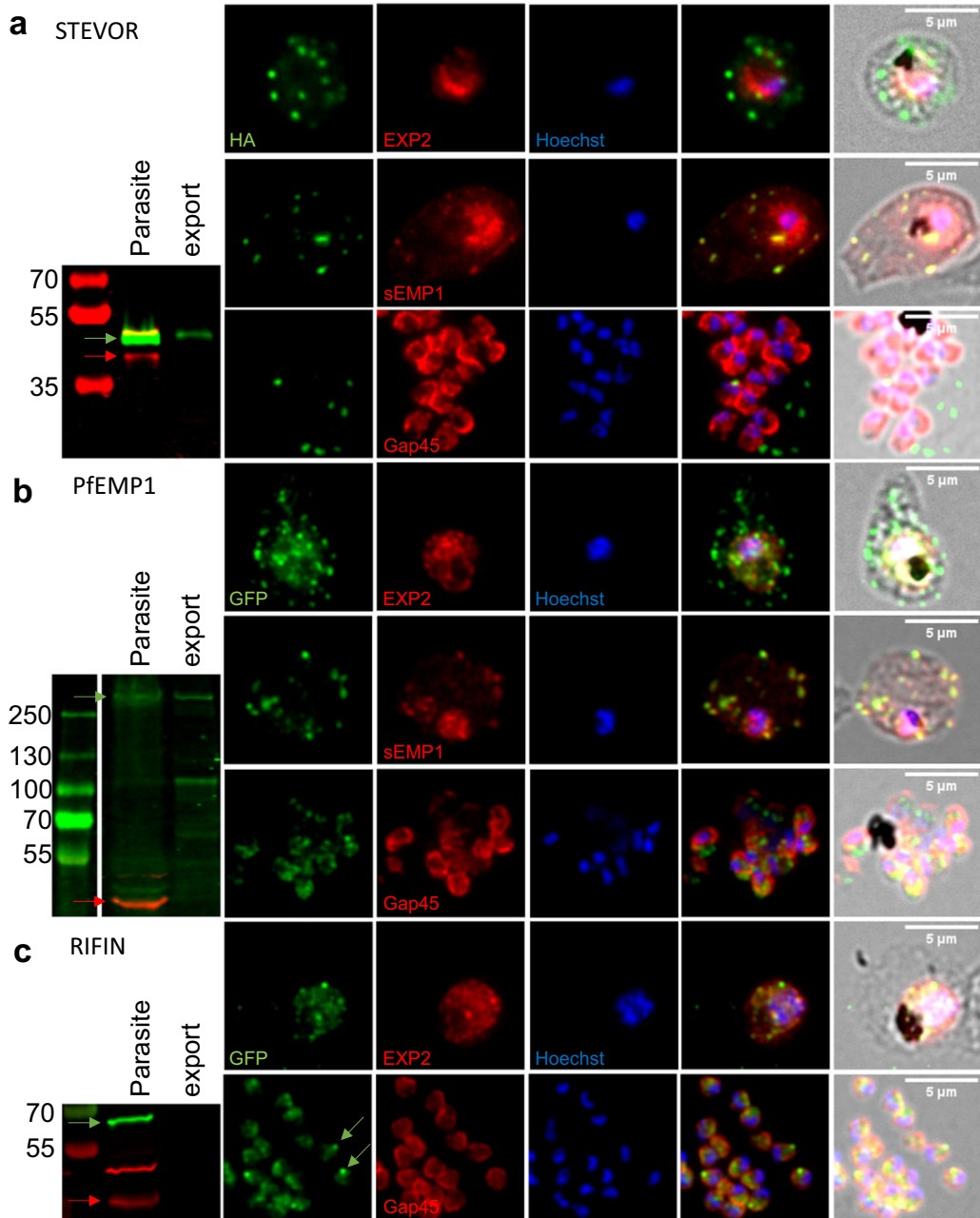

**Fig. 2 Subcellular localization of tagged variant surface antigens.** Protein expression analysis of Pf3D7_1149900.3HA (*stevor*) (**a**), Pf3D7_0421300.GFP (*var*) (**b**) or Pf3D7_1254800.GFP (*rifin*) (**c**) knock-in cell lines. Protein was detected using antibodies against their respective tags. Localization was confirmed using either Aldolase (Parasite internal), EXP2 (PVM), sEMP1 (Maurer's Clefts) or Gap45 (Merozoite surface). For Western blot analysis late stage parasites were enriched by centrifugation on a Percoll gradient. Cells were subsequently lysed using saponin resulting in a parasite and an "export" fraction consisting of the RBC ghost, membranous compartments and soluble proteins within the RBC. Targeted multigene is indicated by green arrow while Aldolase is indicated with red arrow. Exported protein is expected to be seen in the host fraction. DNA was stained using Hoechst33342. **a** The targeted STEVOR is exported to the HCC and located in punctuate dots which co-localize with Maurer's Clefts. **b** The tagged PfEMP1 can be found within the parasite and in the HCC. Exported protein co-localizes with Maurer's Clefts. Protein shows diffuse staining in merozoites. **c** The targeted RIFIN remains entirely within the parasite. It is strongly present within merozoites where it accumulates at the apical end.

analysis (Fig. 3d). RNA was harvested in triplicates grown in the blood of three different donors 18 hpi from all three knock-in cell lines. Relative expression was calculated by normalization to seryl-tRNA synthetase. The expression patterns remained constant when compared to the microarray analysis. None of the multigene families, whether they have been selected for expression or not, showed drastic changes in expression pattern. Despite keeping all three cell lines under the same drug pressure throughout the duration of the experiment, total mRNA levels

remained different between all three multigene families. Relative expression was highest for the knocked in *var* gene, intermediate for the *stevor* and lowest for the *rifin*. Single cell cloning of the *stevor* knock-in cell line in absence of neomycin identified several clones which stopped expression of the fusion protein (Supplementary Figs. 3b, d, 5c). Addition of the drug forced the parasite to resume expression of Pf3D7_1149900 (Supplementary Figs. 3c, 5c), showing that the altered locus is still subject to regular switching events.

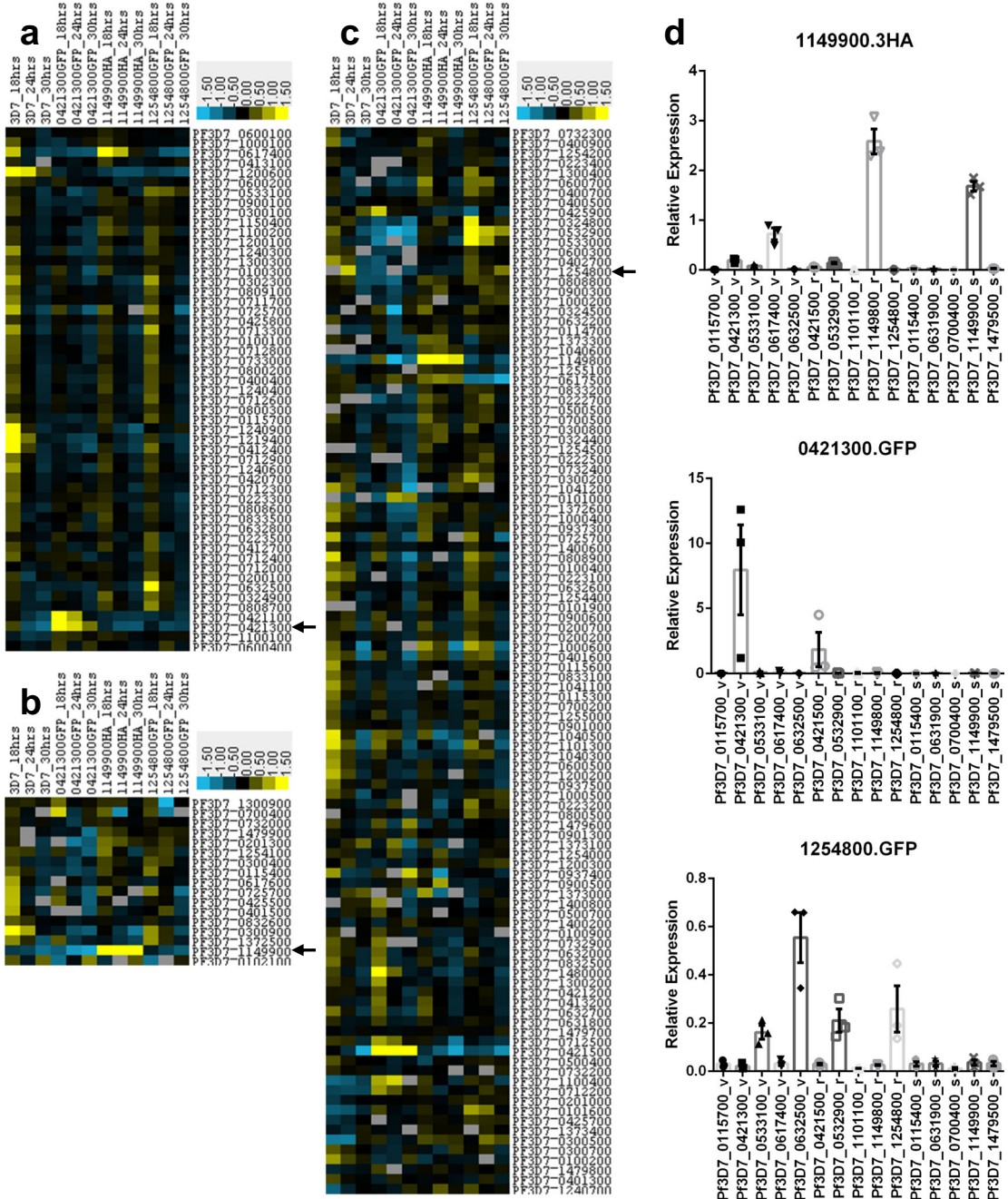

**Fig. 3 Global analysis of P. *falciparum* multigene expression patterns. a–c** Transcriptomic analysis of *var* (**a**), *stevor* (**b**) and *rifin* (**c**) expression after knock-in of one *var* (Pf3D7_0421300), one *stevor* (Pf3D7_1149900) or one *rifin* (Pf3D7_1254800) and compared to parental 3D7 across 3 timepoints (18, 24 and 30 hpi). Blue indicates downregulation, yellow upregulation as compared to refence pool RNA. Data are given as log fold change. Only genes which are either 1.5fold up or downregulated compared to the pool and are detected in at least six samples are shown. Black arrows mark the targeted gene. **a**, **b** Forcing expression of a specific *stevor* or *var* leads to upregulation of that particular transcript and downregulation of all other members. Only one highly expressed *var* gene could be detected after *stevor* knock-in. **c** Forcing expression of a particular *rifin* does not have an impact on overall *rifin* expression. **d** qRT-PCR validation of microarray results. RNA was harvested 6 weeks after the microarray analysis and grown in the blood of three different donors. V, r and s behind the geneID indicate *var*, *rifin* or *stevor* multigene family, respectively. Overall expression pattern remained constant. All data are presented as mean ± standard error of the mean (SEM).

**Activation of natural killer cell inhibitory rifin.** Of the three different *P. falciparum* multigene members studied here, only the *rifin* (PF3D7_1254800) has been previously investigated. This RIFIN has recently been shown to bind to LILRB1, a receptor on the surface of Natural Killer (NK) and B-cells. Binding of RIFINs to this receptor inhibited activation of NK cells in vitro and caused reduced clearance of parasites in NK-cell killing assays[48].

Unfortunately, this study does not show cellular localization of this protein to allow a comparison to our work. In order to verify that the activated genes retained their function, we subjected all three knock-in cell lines and the parental 3D7 to NK-cell killing assays (Fig. 4). In line with the Saito et al.[48] study, activation of the LILRB1 binding RIFIN resulted in a significantly ($p = 0.0203$) reduced NK-cell killing compared to the parental 3D7 while

activation of the specific *var* or *stevor* ($p = 0.4551$ or $p = 01710$, respectively) did not provide protection in these assays.

**Forced expression of specific *yir* in *P. yoelii*.** To evaluate whether the SLI approach can also be utilized for the activation of a single multigene member in vivo we investigated the activation of a single member of the *yir* gene family in the rodent malaria parasite *P. yoelii*. For this we tagged the PLASMED export signal motif containing YIR (PYYM_0017400)[69] with GFP and validated the correct integration by PCR and subsequent sequencing (Supplementary Fig. 4). PLASMED is an export motif in rodent malaria parasites that has been shown to mediate protein export[69].

The correct expression and complete ribosomal cleavage of the PYYM_0017400.GFP fusion protein was established using Western blot analysis of saponin lysed trophozoites (Fig. 5a, Supplementary Fig. 5c). A 63 kDa band corresponding to the full-length PYYM_0017400.GFP fusion protein was detected in both the supernatant and the pellet, indicating that the protein was exported into the HCC. Live fluorescent microscopy indicated that in trophozoite-stage parasites PYYM_0017400.GFP is located

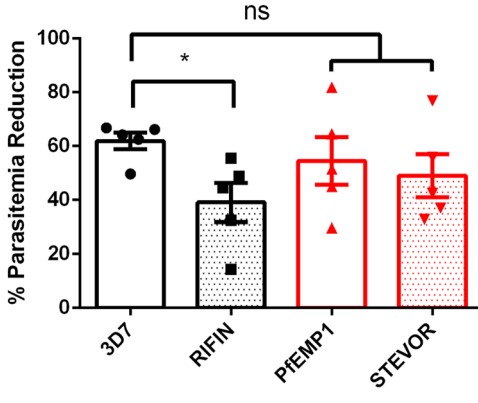

**Fig. 4 Knock-in of the rifin Pf3D7_1254800 protects from NK-cell killing.** The three knock-in cell lines and 3D7 were incubated with human NK cells and parasitemia was monitored over 96 h. Each dot represents NK cells from a separate donor. Paired T-test, *$p < 0.05$.

in the PVM as well as in punctate dots in the HCC while in late schizonts the protein is surrounding individual merozoites (Fig. 5b). Importantly, all the transfected parasites express the GFP tagged YIR, providing a powerful tool to study the biological role of this particular member of the gene family. Our results also clearly indicate that the SLI approach allows the activation of a defined *yir* from its own promoter and are consistent with reports that YIRs have different localization during parasite development[50].

**Transcriptional co-regulation of *yir* genes.** The endogenously tagged PYYM_0017400.GFP cell line now provides an ideal opportunity to study co-regulation of *yir* expression in *P. yoelii*. For this, triplicates of RNA extracted from trophozoite-stage wild-type PYYM as well as 2 clones of PYYM_0017400.GFP were analysed by RNA sequencing. To avoid missing any *yir* genes in our downstream analysis, the RNA seq raw data was mapped to the reference genome of *P. yoelii* 17X due to the larger number of *yir* gene annotation in PlasmoDB (the ID of PYYM_0017400 in the *P. yoelii* 17X annotation is PY17X_1100077). This approach provided with a high confidence transcription data set for the *yir* in *P. yoelii* (Fig. 6). We observed that approximately 50% of the *yir* genes are transcribed within the population, with the majority of the *yir* genes having low expression levels, similar to what was reported by Cunningham et al.[56]. In order to remove *yirs* with low or no expression levels, we omitted *yirs* with estimated counts of less than 20 from our final analysis. The estimated read counts are the counts obtained after the deconvolution process by the program algorithm. It is the estimated number of reads found mapped to the transcript. Using a cut-off value of 20 estimated counts, a total of 60 *yir* genes were found to be commonly expressed in the samples. Comparison of the *yir* expression in parental YM parasites to the two PYYM_0017400.GFP activated clones using Deseq2, identified three distinct groups; four *yir* genes were upregulated, 36 *yir* genes were downregulated and 20 *yir* genes showed no change in expression (Fig. 6a). These results were confirmed by qRT-PCR using seven representative genes with high expression changes from the three groups (Fig. 6b).

Activation of a single *yir* gene led to a statistically significant change in the overall expression pattern of this multigene family. It is particularly striking that four transcripts, the tagged gene PY17X_1100077 and three other *yir* genes (PY17X_0401000,

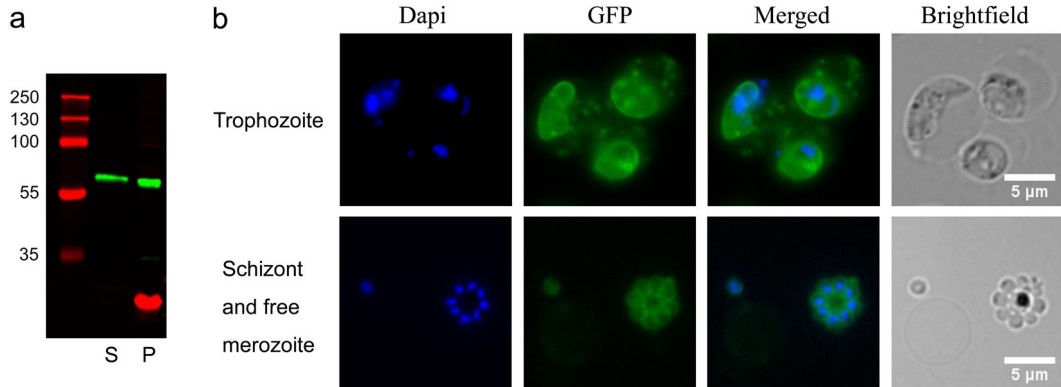

**Fig. 5 GFP fusion protein is expressed in transfected parasites.** Protein expression analysis of PYYM_0017400.GFP knock-in cell line. **a** For Western blot analysis trophozoite-stage parasites were enriched by centrifugation on a Histodenz gradient. Cells were subsequently lysed using saponin resulting in a parasite (P) and supernatant (S) fraction. Exported protein is expected to be seen in the S fraction. A protein of size 63 kDa which correspond to the size of the full-length PYYM_0017400.GFP protein is detected in both supernatant and pellet fraction. The red band corresponds to EXP2 which is a parasitophorous vacuole protein used as a control for the lysis. **b** Live trophozoites observed under florescence microscopy showed PYYM_0017400.GFP localizing in the PVM and in punctate dots. PYYM_0017400.GFP is found to surround individual merozoites in late schizonts and free merozoites. DNA was stained using Hoechst33342.

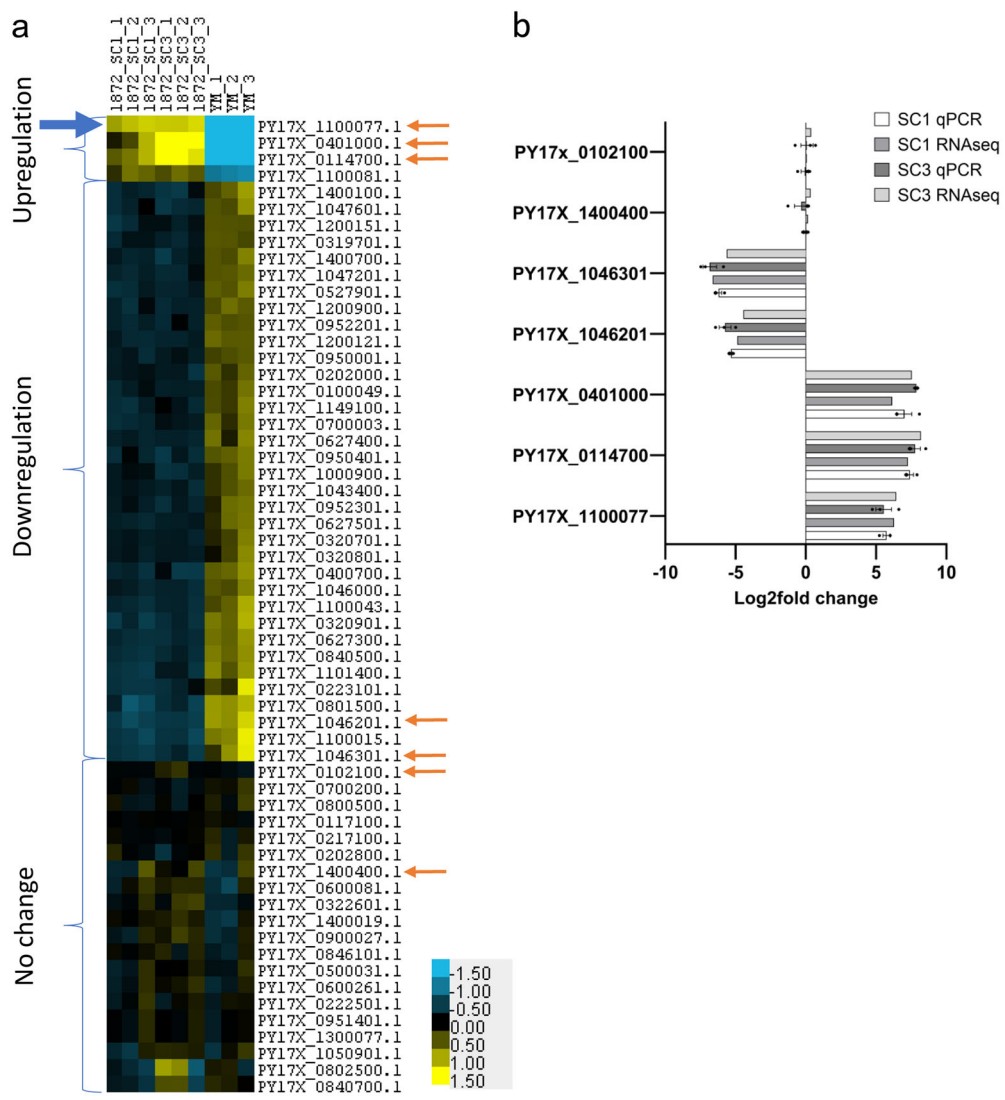

**Fig. 6 Heatmap of *yir* expression compared to that of the parental YM. a** RNA sequencing showed that 60 *yir* genes are expressed (count >20) and are divided into three groups based on the *yir* expression relative to that of the wild-type YM parasites. The blue arrow points to the gene that was tagged in this transfected cell line. Orange arrows point to the genes selected for qRT-PCR validation. **b** Expression of *yirs* in each group were validated using qRT-PCR and results are in line with RNA sequencing results. All data are presented as mean ± standard error of the mean (SEM).

PY17X_0114700, PY17X_1100081) dominate *yir* transcription (for TPM counts see Supplementary Table 1). In addition, 36 other *yirs* were transcriptionally downregulated and another 20 *yirs* were not affected (Fig. 6a). The upregulation of additional *yirs* is similar to what was observed for *var* and *rifin*, with one of the upregulated *yir* gene, PY17X_1100081, being located just downstream of the tagged *yir*, suggesting that open chromatin structure might be contributing to the co-regulation. However, the other two upregulated *yir* genes are located on other chromosomes suggesting a more complex crosstalk between different *yir* loci. Importantly, the data also suggest that there is some level of allelic exclusion as the activation of one *yir* resulted in the transcriptional repression of a substantial number of *yir*. However, unlike *var* gene expression, which is mutually exclusive[20,53], upregulation of a specific *yir* gene affected only a subgroup of *yirs* within the population.

## Discussion

To the best of our knowledge, we believe that this is the first work that describes a novel strategy to study specific members of multigene families. By adapting the previously described SLI method, we were successful in forcing the parasite to express a *stevor*, *rifin*, *var* or *yir* of our choice under its endogenous promoter. Considering the high homology of the C-terminal regions among members of multigene families, careful validation of correct integration is of essence. Generation of tagged cell lines relies on expression of the gene beyond a minimal threshold, potentially excluding genes with low switch rates or defective promoters. We could show that for *stevor* and *var* the targeted gene became the dominant transcript with a concurrent downregulation of other family members. This finding suggests that *stevor*, like *var* genes, are under some mutual exclusive control. Previous studies attempted to address this question by episomally expressing *stevor* exons and evaluating the effect on overall multigene family expression pattern with differing results[38,70]. One obvious shortcoming of these approaches is that they will fail to detect regulatory mechanisms arising from genomic loci not included in the constructs such as enhancer like elements, GC-rich ncRNA or long non-coding RNA arising from introns, all of which can be found in *P. falciparum*[71–73]. Expression of the gene remained constantly high as long as drug pressure was applied to

the parasites. Removal of the drug led to a slow decrease of gene expression in a subpopulation of parasites up to the point where detection by Western blot and qRT-PCR was not possible anymore (parasites were validated approximately 20 cycles after drug removal). The locus was re-activated by addition of the drug to the culture medium. Even though all cell lines were kept under the same drug pressure, relative mRNA levels differed between them. This might be either due to variances in expression timing (RNA was harvested at 18 hpi in all cell lines) or differences in total mRNA transcription. This observation however indicates that the SLI based knock-in of members of multigene families does not alter the expression dynamics but retains the control of the endogenous promoter. Despite the comparatively low expression levels of the *rifin* in Pf3D7_1254800, we were able to reproduce the resistance to NK-cell killing, providing clear evidence that our approach can be utilized for functional studies, even though this multigene family is not under mutual exclusive control. Surprisingly, the RIFIN does not appear to be exported beyond the parasite boundaries, excluding the possibility that the protein interacts with immune cells on the surface of the red blood cell. However, RIFIN have been recently detected within extracellular vesicles from field isolates and it has been shown that *P. falciparum* derived microvesicles can modulate NK-cell activity[74,75]. It is therefore conceivable that the RIFIN is transported to its target cells through parasite derived extracellular vesicles.

Epigenetic regulation in *Plasmodium* species heavily relies on histone modification while DNA methylation appears to play only a very minor role[76]. While most of the parasite genome is euchromatin, heterochromatin stretches from the telomeric ends inwards[12,13]. *P. falciparum* co-expresses genes in genomic proximity based on their stage-specific expression or protein function[77,78]. In this study we observed upregulation of members of multigene families adjacent to our targeted loci. Interestingly, this effect seems to be restricted to genes oriented in a head to tail manner. It could be speculated that the open chromatin necessary for expression of a multigene member stretches for several kilobases both up- and downstream and can lead to the activation of adjacent genes. At this point in time though it cannot be ruled out that this is a result of the introduction of the plasmid into the genomic locus. However, introduction of hDHFR into a subtelomeric region of chromosome 3 did not lead to activation of adjacent *var*[29]. This would suggest that introduction of a drug selectable marker into the vicinity of a *var* alone does not cause expression of that gene. The importance of nuclear organization and non-coding RNA on *var* gene expression has been already described in *Plasmodium falciparum*[79,80]. Considering the importance of enhancer RNA on transcription in higher eukaryotes, we speculate that trans acting enhancers might play a more important role than discovered so far (reviewed in ref. [81]). It is possible that similar mechanisms exist in *Plasmodium* which have not been discovered yet. Furthermore, the importance of non-coding RNA, whether from var introns or GC-rich ncRNAs has already been demonstrated in *P. falciparum*[72,80]. It could be imagined that there are more, so far uncharacterized RNA species that a play a role in the interplay between multigene families.

Our work also provides clear experimental evidence that subgroups of *yir* are, at least to a certain degree, under mutually exclusive control in line with earlier results showing that individual parasites express 1–3 *yirs*[56]. Previous work both in *P. yoelii* as well as *P. chabaudi* had indicated that there are different subgroups of PIR that are regulated differently[56] and have different cellular locations[50] and the SLI linked activation of expression now provides us with the tools to pursue this further. At this stage it is not clear why activation of one multigene member leads to the upregulation of other members of the same gene family or in the case of *P. falciparum* to members of another

multigene family that are not located adjacent to them. While at this stage it cannot be ruled out that this apparent co-regulation is an artefact of the cloning steps involved to generate the parasite lines, the fact that two independent clones in *P. yoelii* show the upregulation of the same *yirs* would argue against this.

The study of multigene families in *Plasmodium* is fraught with challenges. The SLI linked activation approach now provides the community with an alternative tool to rapidly generate homogeneous populations of parasites that express the gene of interest. Despite the need for thorough controls, the ability to activate expression of so far understudied genes will substantially enhance our ability to study the function as well as regulation of different multigene families in *Plasmodium*. Importantly, the approach can be adapted to the study of multigene families in other organisms as well, providing the community with a tool to investigate this challenging problem.

## Methods

**Plasmid construction.** *P. falciparum* knock-in constructs were generated as previously described[62]. In short, ~700 bp of the 3′ end of *var*, *rifin* or *stevor* genes were amplified using KOD Hot start DNA Polymerase (Toyobo) from 3D7 genomic DNA and inserted between the NotI and MluI sites of pSLI-2 × FKBP-GFP or pSLI-2 × FKBP-3HA (cloning between these two sides removes the 2xFKBP domain). For the latter, the 3HA tag was amplified using KOD Hot start DNA Polymerase from a previously generated plasmid and inserted between the BsiWI and SalI sites of pSLI-2 × FKBP-GFP. All constructs were validated by sequencing. The primer sets used for PCR are listed in Supplementary Table 2.

*P. yoelii* knock-in constructs were generated by modifying the existing ePL plasmid which was published previously for use in *P. yoelii* episomal transfection[69] to create an integration plasmid T2A-ePL. DNA corresponding to the last 500 base-pair of PYYM_0017400 gene was generated by chemical gene synthesis (Genscript). We replaced the portion of the ePL which contains (eGFP tag, *P. berghei* EF1 constitutive promoter, the 3′ UTR region of *P. berghei* DHFR/TS gene and the selectable marker *T. gondii* DHFR gene) with an eGFP-T2A-TgDHFR construct. Ligated products were transformed into XL10-gold cells and plated on Ampicillin plates. All positive clones were validated by sequencing. The primer sets used for PCR are listed in Supplementary Table 2.

**P. falciparum culture, transfection and integration.** *P. falciparum* parasites (3D7) were cultured at 37 °C in human erythrocytes under standard conditions[82] with RPMI medium supplemented with 0.5% AlbuMAX II and microaerophilic conditions. Parasites were transfected with 75 μg of purified plasmid DNA (Macherey-Nagel) as previously described[83]. Transfectants were selected with 4 nM WR99210 (Jacobus Pharmaceuticals). Integrants were selected with 400 μg/ml G418 (Promega) as previously described[62]. Genomic DNA was extracted (Macherey-Nagel) and integration PCR was performed as previously described[62]. The primer sets used for PCR are listed in Supplementary Table 3.

**Transfection of Rodent malaria parasites.** Mice infected with wild-type *P. yoelii* YM were used as parasite donor. Mice were euthanized with Valabarb by i.p. injection when the parasitemia reached 15–60%. Infected blood was collected through cardiac puncture with syringe containing heparin solution. Parasitized blood was washed with serum free RPMI (iRPMI) and centrifuged at 2100 rpm with brake 0 for 5 min to remove serum and buffy coat. The RBC pellet was passed through a 50%/60% Histodenz gradient to isolate schizonts. The schizonts were collected and washed once in iRPMI and were allowed to mature with slow shaking in 37 °C for 3 h. Matured schizonts were collected by centrifuging at 2100 rpm for 5 min.

The T2A-ePL vector was linearized with EcoRV and concentrated with ethanol precipitation prior to transfection. 12 μg of linearized T2A-ePL plasmid was used in each transfection. Transfection was carried out by mixing the plasmids with 100ul Nucleofector solution (Lonza) and electroporated with Amaxa Nucleofactor II (Lonza) using program U-33. Transfected parasites were injected intravenously into mice and transfectants were selected with a 3 day on and off regime of pyrimethamine (MD laboratories) injections. Integration was confirmed by PCR and southern blotting. Integration was confirmed by PCR and southern blotting. The primer sets used for PCR and for the southern blot probe are listed in Supplementary Table 3.

**Imaging.** Thin blood smears of synchronous parasite cultures (intracellular stages) or Percoll-enriched schizonts (merozoites)(MP Biomedicals) were fixed either with ice cold methanol for 10 min (intracellular stages) or airdried (merozoites). Airdried slides were fixed with 4% paraformaldehyde (Sigma) for 5 min at room temperature followed by membrane permeabilization using 0.01% saponin (Sigma) for 5 min at room temperature. Slides were blocked with 1% bovine serum albumin (Sigma) in PBS with 1% Tween-20 (Sigma). Primary antibodies (rabbit-anti EXP2

1:500; rabbit-anti sEMP1 1:500; rabbit-anti Gap45 1:50000; mouse-anti GFP (Roche) 1:200; rat-anti HA (Roche) 1:500) were incubated for 1 h at room temperature in blocking buffer. After two washes with PBS with 0.1% Tween-20, appropriate secondary antibodies (Alexa 594 anti-rabbit, Alexa 594 anti-rat, Alexa 488 anti-rabbit 1:1000 (all from Jackson ImmunoResearch)) and Hoechst 33342 (1 μM) were incubated for 1 h at room temperature. After another two washes slides were mounted with Fluoromount G (SoutherBiotech). Slides were visualized on a Nikon Eclipse Ti fluorescent microscope with a Nikon Plan Apochromat Lambda 100X Oil objective. Pictures were taken using an Andor Zyla sCMOS camera and analysed using ImageJ 1.52 h. For live cell imaging of transgenic *P. yoelii* parasites with GFP tagged YIR, parasites were incubated with DAPI (1 ug/ml in iRPMI) for 10 min to stain the nuclei. The cells were then viewed in between slide and coverslip as described above.

**Parasite fractionation, trypsin shaving and Western Blot analysis.** Trophozoite-stage parasites (38 hpi) were enriched on 68% Percoll (MP Biomedicals). Parasites were separated from the host red blood cell by incubation with 0.05% saponin (Sigma) in iRPMI for 5 min at room temperature followed by centrifugation for 5 min at 900*g*. The parasite pellet was resuspended in iRPMI. Both fractions boiled at 95 °C for 5 min after addition of 6x Laemmli loading dye. Tryptic digest of surface exposed proteins was performed as previously described[46]. In short, percoll-enriched parasites were incubated for 10 min with TPCK treated Trypsin (Sigma) at 37 °C in iRPMI. Reaction was stopped by addition of malaria culture medium supplemented with 1 mM Phenylmethylsulfonyl fluoride (Sigma) and protease inhibitor cocktail (Nacalai Tesque). Proteins were sequentially extracted with 1% Triton X-100 (Sigma) and 2% SDS. Parasite proteins were separated in 12% SDS-polyacrylamide gels and transferred to low fluorescence PVDF membranes (BioRad). Immunoblots were blocked with Odyssey Blocking Buffer (PBS base) (LiCor) for 1 h at room temperature. Primary antibodies (Aldolase 1:3500; GFP 1:1000; HA 1:1000) were incubated either 1 h at room temperature or over night at 4 °C in blocking buffer. After three washes with PBS and 0.1% Tween-20 (Sigma), appropriate secondary antibodies (IRDye680rd anti-rabbit 1:10000; IRDye800cw anti-mouse 1:30000; IRDye800cw anti-rat 1:30000 (all from LiCor)) were incubated for 1 h at room temperature in blocking buffer with final concentrations of 0.1% Tween-20 (Sigma) and 0.05% Sodium dodecyl sulfate (1st Base). Membrane was washed three more times with washing buffer and once with PBS. Membranes were scanned with a LiCor Odyssey CLx imager.

**Microarray analysis.** Total RNA was isolated from highly synchronized *P. falciparum* ring- and trophozoite-stage parasites (18, 24 and 30 hpi) and hybridized against a reference pool of 3D7 cultures as previously described[77].

**P. yoelii RNA extraction and cDNA synthesis.** Mice were inoculated intravenously with parasites in the ring stages and harvested 30 h post infection to get trophozoite enriched parasites. Mice were bleed and whole blood was collected and passed through a non-woven fabric (NWF) filter (ZhiXing Bio S&T) to remove leucocytes. The filtrate was then centrifuged at 2200 rpm for 5 min (acc5, dec0) and washed 2 times with iRPMI. 200ul of erythrocyte pellet was added to 2 ml Trizol (prewarmed to 37 °C), inverted a few times to get rid of cell clumps and incubated at 37 °C for 10 min. The Trizol-erythrocyte mixture was then frozen in dry ice and kept in −80 °C until ready for RNA extraction. RNA extraction was carried out as previously described[84]. The extracted RNA was purified using RNA cleanup kit following manufacturer's protocol (Qiagen) and stored in −80 °C until further analysis.

Reverse transcription of RNA was performed with SuperScript III reverse transcriptase according to manufacturer protocol.

**RNA sequencing and analysis.** Triplicates of each RNA sample were outsourced to the genome facility at the Singapore Centre for Environmental Life Sciences Engineering (SCELSE) for RNA sequencing. RNA library preparation was performed according to Illumina's TruSeq Stranded mRNA protocol with 1000 ng of total RNA as input. The PCR amplification step, which selectively enriches for library fragments that have adapters ligated on both ends, was performed according to the manufacturer's recommendation but the number of amplification cycles was reduced to 12. Each library was barcoded with one of Illumina's TruSeq LT RNA barcodes to enable library pooling for sequencing.

Finished libraries were quantitated using Promega's QuantiFluor dsDNA assay and the average library size was determined on an Agilent Tapestation 4200. Library concentrations were then normalized to 4 nM and validated by qPCR on a QuantStudio-3 real-time PCR system (Applied Biosystems), using the Kapa library quantification kit for Illumina platforms (Kapa Biosystems). DNA libraries were then pooled at equimolar concentrations and sequenced on an Illumina MiSeq sequencer at a read-length of 150 bp paired-end.Removal of Illumina adapters was made with the use of Trimmomatic[85] and Kallisto was used to create an index file from 17X1.1 CDS (extracted from PlasmoDB on 28 May 2018) for quantification. We performed the RNAseq analysis using Kallisto to map the reads against the annotated PY17X1.1 index file with boostrap 100 and thread 4 to obtain the transcript per million (TPM) count. Differential gene analysis was done by sending the TPM from Kallisto into DEseq2 software.

**Whole genome sequencing and analysis.** Genomic DNA of two clones of all three *P. falciparum* cell lines was extracted using phenol:chloroform:isoamylalcohol (25:24:1, Invitrogen) followed by ethanol precipitation as per manufacturers protocol. All following steps were performed by the Genome Institute of Singapore. DNA libraries were prepared by 1D native, PCR free barcode ligation. Multiplexed libraries were sequenced on an Oxford Nanopore GridION platform release 19.12.6 on a single flowcell (Basecalling/mode: Guppy 3.2.10/High accuracy). *P. falciparum 3D7* reference genomes were extracted from PlasmoDB and edited to introduce the appropriate plasmid sequence at the gene of reference, resulting in three different edited genomes. Reads were demultiplexed and mapped to the according edited reference genome using minimap2 and sorted with Samtools. Mapped reads were visualized using IGV (Version 2.8.2). In order to quantify correct integration events, 200 bps of both ends from every read containing the plasmid were mapped against the reference genome. Mapped loci were counted and percentage of correct integration events calculated.

**Real-time PCR.** Unique primer pairs to subsets of *var*, *stevor*, *rifin* and *pir* genes were designed and their efficacy tested against either *P. falciparum* or *P. yoelii* genomic DNA. RNA extraction and cDNA synthesis were performed as mentioned above. PCR reactions were performed using qPCRBIO SyGreen Mix Lo-ROX (*P. falciparum*; PCR Biosystems) or SYBR Green qPCR mastermix (*P. yoelii*; Invitrogen) according to manufacturers protocol and analysed using an Applied Biosystems 7500 Fast Real-Time PCR System. The primer sets used for quantitative Real-time PCR are listed in Supplementary Table 4.

**Natural killer cell killing assay.** NK cell killing assay was performed as previously described[75]. Briefly, NK cells (CD56⁺CD3⁻) were obtained from PBMC or patient buffy coat by negative selection using EasySep™ Human NK Cell Enrichment Kit (Stemcell Technologies). Synchronized schizonts-stage iRBCs at a parasitemia of 0.5% were incubated with NK cells at a ratio of 1:10 for 96 h in malaria culture medium. Parasitemia was quantified by flow cytometry and Giemsa-staining. Parasite DNA was stained using Hoechst 33342 (20 μg/ml) (Thermo Scientific). Calculation for reduction in parasitemia is as follows:

$$\% \text{ Parasitemia reduction} = \frac{\text{Parasitemia}_{iRBC} - \text{Parasitemia}_{NK+iRBC}}{\text{Parasitemia}_{iRBC}} \times 100.$$

**Statistics and reproducibility.** Statistical analysis of all experiments was performed with GraphPad PRISM 6.0 software (GraphPad Software. Inc., CA, USA). Statistical methods, sample sizes and probability values used to analyze relevant experiments are described in the figure legends. *Plasmodium falciparum* parasites were grown in blood of different human donors, *Plasmodium yoelii* parasites in independent mice to generate biological replicates. Expression levels of *P. falciparum* multigene families was calculated as previously described[86].

**Ethics statement.** This study was carried out in strict accordance with the recommendations of the NACLAR (National Advisory Committee for Laboratory Animal Research) guidelines under the Animal & Birds (Care and Use of Animals for Scientific Purposes) Rules of Singapore. The protocol was approved by the Institutional Animal Care and Use Committee (IACUC) of the Nanyang Technological University of Singapore (Approval number: ARF SBS/NIE-A-0379). All efforts were made to minimize the suffering.

**Reporting summary.** Further information on research design is available in the Nature Research Reporting Summary linked to this article.

## Data availability

All relevant data are within the paper and its Supporting Information files are available from the authors upon request. The microarray and RNA sequencing data discussed in this publication have been deposited in NCBI's Gene Expression Omnibus[87] and are accessible through GEO Series accession number GSE128123 and GSE152162, respectively. The whole genome sequencing data have been deposited with links to BioProject accession number PRJNA637985 in the NCBI BioProject database (http://www.ncbi.nlm.nih.gov/bioproject/637985). Source data underlying the plots shown in figures are provided in Supplementary Data 1.

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

## Acknowledgements

The authors would like to thank Professor Marek Mutwil and Dr Devendra Shivhare for their valuable input in the Kallisto workflow. We would like to thank Anthony Holder for his kind gift of the Gap45 antibody. Special thanks to Tobias Spielmann for sharing of the plasmids and information about the SLI system before publication. Han Ping Loh was funded by A*STAR Graduate Academy (AGA). Radoslaw Igor Omelianczyk was supported by SINGA. This work was funded by the Singapore Ministry of Health's National Medical Research Council under its Cooperative Basic Research Grant (NMRC/CBRG/0040/2013) and Open Fund Individual Research Grant (NMRC/OFIRG/0058/2017). M.C. J.C. and P.R.P. were partially supported by the National Research Foundation of Singapore through the Singapore–MIT Alliance for Research and Technology's (SMART) Interdisciplinary Research Group in Infectious Disease and Anti-Microbial Resistance Research Programs.

## Author contributions

Conceptualization, P.R.P.; Methodology, R.I.O., H.P.L and M.C.; Investigation, R.I.O. and H.P.L.; Bioinformatic Analysis, R.I.O., H.P.L., S.B. and R.H.; Writing—Original Draft, P.R.P., R.I.O., H.P.L; Writing—Review & Editing, P.R.P., R.I.O., H.P.L., J.C.; Funding Acquisition, P.R.P.

## Competing interests

The authors declare no competing interests.
