## [Peer Review File · Communications Biology]

Reviewers' comments:

Reviewer #1 (Remarks to the Author):

Comments on the paper "Rapid activation of distinct member of multigene families in Plasmodium spp." by Omeliansczyk and colleagues.

The work by Omeliansczyk and colleagues describes the application of the SLI vector system on variant gene family members of two Plasmodium species. While two knockins lead to ambiguous (var) or hard to interpret (rif) results, the latter based on the yet unknown biology of rif transcription, the experiments dealing with the third gene family member (stevor) seems sound. In my opinion the manuscript needs additional, in part fundamental, experimental and formal modification to become acceptable.

General comments:

The manuscript by Omeliansczyk and colleagues employs the SLI principle to tag members of variant gene families of two Plasmodium species, *P. falciparum* and *P. yoelii*. The idea of the approach is very reasonable, however, the results are not yet sufficiently explored and a couple of questions arise regarding the efficiency of the system, especially for *P. falciparum* var and rif genes.

The authors selected determined var rif and stevor genes in order to tag them and show the proof of principle. While the selection of the rif target was clear, it is puzzling why the var gene of the C-type was selected. The determined gene was found expressed in one study which tried to associate binding phenotypes to differently transfected CHO cell lines (Golnitz et al., Malaria Journal 7:14, 2008), but even in that study – due to two upregulated var genes as happened in the manuscript – no very affirmative conclusion could be drawn regarding receptor binding specificity. Given the importance of var genes in pathogenic processes, the authors should have tried to target a var gene with established phenotype when expressed, such as var2csa, or a DC8/13 containing var such as PF3D7_0400400 (EPCR binding) or PF3D7_0425800 (ICAM1 binding). After establishing the knockin into the var locus, the authors later observed that an adjacent var gene (Pf3D7_0421100, microarray) is also stronger transcribed. The explanation given is that chromatin is somehow activated/opened by their knockin/activation. This would somehow justify the activation of an upstream (head to head) localized rif gene (Pf3D7_0421500), since chromatin modifications are most decisive at 5' ups regions and the var promoter may be bidirectional. Since the co-activated var gene Pf3D7_0421100 is intercalated with a non-activated rif gene Pf3D7_0421200 (head to head to var Pf3D7_0421100), the hypothesis "opened chromatin" does not stand and complicates a simplistic model of general chromatin activation at the inserted locus. In addition, why was the co-activated var Pf3D7_0421100 not also tested in RT-qPCR (Fig 2d)? Regarding the design of the figure 2d, it would be easier for the reader if Figure 2 d names clearly which gene is rif, var and stevor by labeling the gene IDs with v, r and s, or maybe order them in the plot for each gene type. Please add "+" and "-" to the heatmap legend of genes.

In my opinion, if the authors can't show that tagging and posterior selection of one var gene leads to the unique expression of this var gene, the system is not yet very helpful to elucidate later protein function or receptor specificity.

The pattern of PfEMP is somehow unusual showing the very distinct punctuate pattern instead of a green ringform-like appearance, and it resembles the localization of SBP1 (Saridaki et al. 2009, Traffic 10, 2) which is on the inside of IRBC, under the RBC membrane. Although not absolutely required, to elucidate the question if the tagged PfEMP1 is really localized externally, the authors may conduct mild trypsin digestion previous to Western blots with IRBC membrane fractions as done by others. Is it possible that the use of only Albumax 1 in the medium has to do with the

perhaps imperfect translocation of PfEMP1?

It is known that the active var locus is transcribed most actively in early ring stage parasites and normally ceases after 15-18 h after reinvasion (Kyes et al. 2007, Mol Microbiology 64, 4). Regarding this, the harvest of RNA for var transcription would be ideally at an additional time point at 10-15 h p.i., but since the experiment worked out this not a major problem.

Fig 3: Is the difference between the RIFIN-tagged strain and the var or stevor-tagged strain significant? The authors should inform this. In the fluorescence microscopy, the authors show that tagged RIFIN is not exported although it is a rif A type gene (which are exported RIFINs, following Mats Wahlgren's group – Joannin et al. 2008 BMC Genomics). Is it possible that the GFP poses a steric hindrance for export? Then, if it is not exported, is it possible that by chance another A type RIFIN does the observed inhibition? In this sense, an elegant way to show if the expressed RIFIN is functional would be to incubate the tagged RIFIN-expressing strain in the presence of rapalog which should mislocate RIFIN away and therefore abolish the observed effect. If this does not happen, another expressed A RIFIN would exert the observed effect (which is possible anyway since rif transcription/expression is not monoallelic).

In my opinion, the stevor results are convincing and need no further work. The yir gene experiments also appear sound.

As a general question, why wasn't the functionality of the "knock sideways" cassette not tried (mislocating tagged proteins via the FKBP/FRB domains)?

Minor/formal aspects:

The expression "multigenes" in the discussion section is very unlucky and should be changed. The authors comment that parasites selected to transcribe/express one specific stevor gene ceased to do so during generations without selection. Can this be shown also by Western blot or fluorescence microscopy? In general, please revise the whole text for typographic mistakes starting in the title changing to "distinct members". In a couple of places "Plasmodium" is written in small letters. In the introduction you mention small VSAs when you indeed mean the pir protein family, please do so. The last sentence in the introduction is too long and should be reformulated.

Abbreviations should be written out at their first use such as PFA (probably paraformaldehyde).

PLASMED in the Results section is also puzzling for the normal reader as is SCELSE (apparently a facility). In Figure 1, I find the classification "host" misleading and would rather prefer "IRBC membranes" or "IRBC ghosts". Also, the reader understands the

message of Figure 1 easier if on the right side of the IFA pics the examined protein species is named, such "PfEMP1", "RIFIN" and "STEVOR". Please use also the same letter type in the figures and not mixtures of Arial, Times and Calibri.

Please reformulate the Methods section in a way that it looks less like a lab journal. As an example, the description of the reverse transcription looks very much like a lab journal recipe. Although I believe that the primers used for qPCR were designed appropriately, the authors should invest a little more time and effort in describing how they were designed, and which sequence these had or alternatively a reference may be given.

Regarding the supplementary material, please provide a plasmid map with oligo localization so that the reader can monitor what proves integration and what hints to the presence of residual episomal copies in the transfectants.

Reviewer #2 (Remarks to the Author):

Omelianczyk et al. have developed a transfection strategy that allows for the activation of a defined member of a multi gene family, thus making this gene amenable to molecular and functional analysis. The authors have tested their approach for a member of the *P. falciparum* var, stevor and rifin gene family each and the *P. yoelii* yir family. They then investigated the question of how the forced activation of a gene affected expression of other family members and whether or not there is a cross-talk between different multi-gene families. Microarray data presented suggest a nuanced picture depending on the gene family investigated. Moreover, initial evidence is presented in support of a cross-talk between multi-gene families. Comparable data were obtained for the *P. yoelii* yir family. The authors further demonstrate proof-of-principle for their claim that the new transfection technology can be used for functional studies. The manuscript is interesting and without doubt will stimulate further studies on the mechanisms underpinning regulation of multi-gene family in Plasmodium.

Comments:

1. A cartoon outlining the transfection strategy would help the reader navigate the manuscript.
2. What are the evidences that tagging the proteins does not affect their subcellular localization? Is the tagged PfEMP1 presented on the surface of the parasitized erythrocyte?
3. It is unclear whether the activation of genes in close proximity to the integration site is a natural event, such that is it part of a biological mechanism, or whether it is an artefact resulting from the integration event and the forced gene activation. The activation of a second var gene in close proximity to the tagged gene would argue in favor of the latter hypothesis. A control might be to tag an unregulated gene adjacent to a var gene and see if this var gene is also activated.
4. The data presented suggest a cross-talk between different multi-gene families. Can the authors speculate about possible mechanisms?

Reviewer #3 (Remarks to the Author):

This manuscript uses a recently developed method called Selection Linked Integration (SLI) to study selected genes in multi-gene families in *P. falciparum* (var, rifin, and stevor) and in *P. yoelii* (yir). Gene expression was induced (forced) by drug pressure, and the expression levels of the genes in these families were evaluated using microarray, RNA-seq, qPCR and antibodies against HA and GFP tags. Studying gene families in Plasmodium parasites have been difficult, and this method appears to be a good approach for investigating the roles of these genes in parasite gene expression and development. Interesting information on mutual exclusive gene expression, co-regulation and interaction of a specific rifin gene with host cells were obtained.

Although the plasmids and the strategy have been described previously, some introduction on the original SLI method will help readers who do not know the method. Additionally, on page 4, second paragraph, the authors stated that "we have developed..." leading to thinking of a new approach developed by this work. Again, page 4, a plasmid map should be cited/provided when introducing the "resistant cassette".

Some minor points:

- 1, Page 5, line 3, PFA: spell out.
- 2, Page 6, figure 1: Also label each panel of the images with STEVOR, pfEMP1, and RIFIN, respectively.
- P3, age 7, second para, line 3: remove `,' after (Pf3D7_1149900).
- 4, Figure 2a: It would help if the parasite groups (top) are also labeled.
- 5, Page 10, line 11: "var or stevor".
- 6, Page 11, second paragraph, line 4: Replace "was analysed" with "were analysed".
- 7, Page 11, second paragraph, line 9, "20 estimated counts": Please explain. Line 2 from the

bottom: Figure 4a should be Figure 5a. Expression data from only 60 yir genes are presented. What are the expression levels for other genes in the genome?
8, Figure 5b: Are the data adjusted with expression levels from YM?
9, Page 14, line 1: iRPMI? Spell out.
10, Page 14, last paragraph, Trophozoites staged parasites.
11, Page 17, line 5: PBS and 0.1% Tween-20.
12, Page 17, third paragraph, NFW filter: Please provide company name.
13, Page 19, line 3, Omelianczyk was supported by..?
14, Page 22, Supp. Table 1, TPM: spell out.

Reviewer #4 (Remarks to the Author):

The manuscript by Omelianczyk et al addresses an important issue, which is the lack of tools to efficiently study the regulation of variant surface antigens (VSA) of different Plasmodium species. The authors used a selection linked integration approach to develop transgenic parasite lines of *P. falciparum* and *P. yoelii* in which single members of the VSA families var, rifin, stevor or yir are forced to be expressed. Subsequently, the parasite lines were analysed by immunofluorescence assays and RNAseq to determine the effect of forced expression of a single VSA variant on other members of the VSA gene families. While this is a nice concept to learn more about the co-regulation or mutually exclusive expression of the different VSA families, I have concerns regarding the integration efficiency of the transgenes in the parasite lines (in particular of the rifin line). As this has a critical impact on the interpretation of all of the results, the authors should better characterize the parasite lines to proof that the transgenes are indeed expressed on a population level.

Major points:

1. Figure S1: In this integration PCR screen, a wild type band is still evident for all of the transgenic parasite lines. The authors interpret this band as irrelevant because it may represent other VSA genes, which may be amplified due to the high sequence homology around rifins, stevors, and var genes. However they haven't shown that this is really the case. In my experience (and to my frustration), wild type DNA often persist after SLI selection with G418, even for single copy genes. Thus, cloning by limiting dilution of the parasite lines is still required in most cases. Therefore, the authors should use a different strategy to convincingly show integration of the transgene in the majority of the population, for example by FACS analysis, Southern blot or immunofluorescence analysis of the population (what proportion of parasites express the tagged variant?).
2. Figure 1 Western Blot: I am confused by the statement that RIFINs and STEVORs are expected both in the pellet and SN fraction after saponin treatment (Results, section 1, paragraph 2). According to many publications, saponin fractionation under the conditions used by the authors leads to the separation of (infected) erythrocyte cytosol and soluble parasitophorous vacuole content from a pellet, which contains the parasite wrapped in the permeabilized host cell membranes. So RIFINs and STEVORs, which are transmembrane proteins throughout their export pathway, would be expected in the pellet. This is indeed consistent with the data shown in Figure 1. As a control the authors should run a WB for example on EXP2 or sEMP1 or some other Maurer's cleft associated protein e.g. SBP1. Importantly, their conclusions about export of STEVOR and PfEMP1, and of parasite-restricted expression of RIFIN is still validated by the IFA images.
3. The authors claim that transgenic stevor expression results in downregulation of all other stevor- but in fact there are several stevor variants that are upregulated in comparison to the 3D7 control according to Figure 2A (e.g. Pf3D7_0201300). Please comment on this. In my understanding this argues against mutually exclusive expression. However, this might also be a result of incomplete integration of the transgene at the stevor locus across the parasite population.
4. Figure 3: According to the data shown in figure 2, the GFP tagged RIFIN variant is not

upregulated relative to the 3D7 wild type control. Still, reduction of NK cell killing is only observed in the RIFIN transgenic parasite line, in which, however, the GFP tagged variant is not exported from the parasite. How do the authors explain these observations?

5. Figure S2: Although there is clear evidence for integration, controls are missing that show that the wild type locus has been deleted from the parasite population and that equal amounts of DNA are analyzed (by PCR of an unmodified locus). This is critical for the interpretation of the gene expression results, as any parasites which are not subject to forced *yir* expression may express different dominant VSA variants.

6. Please include in the discussion other publications addressing the impact of sVSA promoter activation on the regulation of other VSA family members, e.g. Witmer et al 2012, Howitt et al 2009.

Minor issues:

1. Introduction, line 1: The malaria parasite exports...

2. Figure 2a: please clarify what kind of data are shown in the heat map –(log) fold changes or expression levels?

3. Figure 2a: throughout the paper, the order of transgenic parasite lines is Stevor – var- rifin, but in this Figure the order is var – stevor – rifin. For consistency and clarity it would be helpful to change the order in the heatmap.

4. Figure 2d: It is unclear in this Figure which of the VSAs analyzed by qPCR belong to which family, please label them.

5. Figure 2d: Which tRNA synthetase was used for normalization?

6. Page 8: Rifin Pf3D7_0421200 is oriented tail to tail, not head to head, with var Pf3D7_0421300. Rifin Pf3D7_0421500 is not directly adjacent to var Pf3D7_0421300 but there is a RUF6 ncRNA gene (Pf3D7_0421200) in between.

7. Page 10 line 11: correct var or stevor

8. Page 10 2nd paragraph: what is PLASMED - should this be plasmid? Otherwise please explain.

9. Materials and Methods: NK cell killing assay – what media was used? "x 100" is missing in the formula

10. Primer sequences used for cloning/ qPCR/ diagnostic PCR should be provided as a supplement.

Reviewer 1:

“While the selection of the rif target was clear, it is puzzling why the var gene of the C-type was selected. The determined gene was found expressed in one study which tried to associate binding phenotypes to differently transfected CHO cell lines (Golnitz et al., Malaria Journal 7:14, 2008), but even in that study – due to two upregulated var genes as happened in the manuscript – no very affirmative conclusion could be drawn regarding receptor binding specificity. Given the importance of var genes in pathogenic processes, the authors should have tried to target a var gene with established phenotype when expressed, such as var2csa, or a DC8/13 containing var such as PF3D7_0400400 (EPCR binding) or PF3D7_0425800 (ICAM1 binding).”

The approach presented here was specifically developed to activate genes which have not been studied before. Therefore, both the var as well as the stevor gene were selected as they lacked a previously described binding phenotype.

“After establishing the knockin into the var locus, the authors later observed that an adjacent var gene (Pf3D7_0421100, microarray) is also stronger transcribed. The explanation given is that chromatin is somehow activated/opened by their knockin/activation. This would somehow justify the activation of an upstream (head to head) localized rif gene (Pf3D7_0421500), since chromatin modifications are most decisive at 5' ups regions and the var promoter may be bidirectional. Since the co-activated var gene Pf3D7_0421100 is intercalated with a non-activated rif gene Pf3D7_0421200 (head to head to var Pf3D7_0421100), the hypothesis “opened chromatin” does not stand and complicates a simplistic model of general chromatin activation at the inserted locus.”

In our open chromatin explanation we propose that activation of a multigene could lead to the upregulation of adjacent multigenes which are in a head to tail orientation but not those in a head to head orientation. Activation of var 0421300 leads to upregulation of var 0421100 and rif 0421500 (which are both in a head to tail orientation, but not rif 0421200. We changed the wording in the manuscript to make it more understandable and added an overview of the genomic region from PlasmoDB to the supplements (supplementary figure 2a).

“It is though interesting to note that the activation of var (Pf3D7_0421300) led to the concurrent activation of the adjacent rif (Pf3D7_0421500). The genes are oriented in a head to tail manner and separated by about 5.1kbps which include the non-coding RNA RUF6. Another rif (Pf3D7_0421200), which is located 800bps downstream of the targeted var in a tail to tail fashion was apparently not affected. A similar activation of an adjacent rif (Pf3D7_1149800) was also seen in the case of the stevor-activated parasite line (Figure 3c).”

The previously mentioned paper (Golnitz et al., Malaria Journal 7:14, 2008) makes a similar observation where panning on various CHO cell lines causes high expression of var 0420900 and 0421100. In these selected cell lines var 0421300 transcript could also be detected in most of the cell lines which were high for 04210900 and 0421100 expression.

“In addition, why was the co-activated var Pf3D7_0421100 not also tested in RT-qPCR (Fig 2d)?”

We assumed that testing for rif Pf3D7_0421500 would be sufficient. We have now performed the requested qPCR reaction as well as expression of the non-regulated rif Pf3D7_0421200 and both are in line with the microarray data (supplementary figure 2b).

“Regarding the design of the figure 2d, it would be easier for the reader if Figure 2 d names clearly which gene is rif, var and stevor by labelling the gene IDs with v, r and s, or maybe order them in the plot for each gene type. Please add “+” and “-” to the heatmap legend of genes.”

We agree. The figure (now figure 3) was modified accordingly.

“The pattern of PfEMP is somehow unusual showing the very distinct punctuate pattern instead of a green ringform-like appearance, and it resembles the localization of SBP1 (Saridaki et al. 2009, Traffic 10, 2) which is on the inside of IRBC, under the RBC membrane. Although not absolutely required, to elucidate the question if the tagged PfEMP1 is really localized externally, the authors may conduct mild trypsin digestion previous to Western blots with IRBC membrane fractions as done by others”

The pattern we see is highly similar to that described by for the GFP tagged PfEMP1 (Melcher et al. 2010, Cellular Microbiology) which has been shown to be exposed to the surface. We performed the surface trypsinization experiment using a slightly adapted version of the protocol described in Baruch et al., 1996, PNAS 93 and show sensitivity of the tagged PfEMP1 to trypsin digestion (supplementary figure 3a).

“Is it possible that the use of only Albumax 1 in the medium has to do with the perhaps imperfect translocation of PfEMP1?”

This is a typographical error in the methods section as Albumax 2 was used throughout the experiments. This has been corrected.

“It is known that the active var locus is transcribed most actively in early ring stage parasites and normally ceases after 15-18 h after reinvasion (Kyes et al. 2007, Mol Microbiology 64, 4). Regarding this, the harvest of RNA for var transcription would be ideally at an additional time point at 10-15 h p.i., but since the experiment worked out this not a major problem.”

The time points were selected to cover peak expression times of all three studied multigene families.

“Is the difference between the RIFIN-tagged strain and the var or stevor-tagged strain significant?”

The difference between the RIFIN tagged cell line and 3D7 / var tagged strain is significant

“In the fluorescence microscopy, the authors show that tagged RIFIN is not exported although it is a rif A type gene (which are exported RIFINs, following Mats Wahlgren’s group – Joannin et al. 2008 BMC Genomics). Is it possible that the GFP poses a steric hindrance for export? Then, if it is not exported, is it possible that by chance another A type RIFIN does the observed inhibition?”

To our hands a GFP tag does not normally cause mislocalization something that has also been shown in the previously mentioned study by Melcher et al. Independent of localization we can imagine other ways how the RIFIN might cause NK cell inhibition. RIFINs have been recently detected in extracellular vesicles from field isolates (Abdi et al. 2017 Wellcome Open Res). Furthermore, we have previously shown that the microvesicles exported by the parasite are able to modulate NK cell activity (Ye et al., 2018 PLOS Pathogens). We have modified the text to indicate this possibility.

“Surprisingly, the RIFIN does not appear to be exported beyond the parasite boundaries, excluding the possibility that the protein interacts with immune cells on the surface of the red blood cell. However, RIFIN have been recently detected within extracellular vesicles from field isolates and it has been shown that *P. falciparum* derived microvesicles can modulate NK-cell activity [76, 77]. It is therefore

conceivable that the RIFIN is transported to its target cells through parasite derived extracellular vesicles.”

“In this sense, an elegant way to show if the expressed RIFIN is functional would be to incubate the tagged RIFIN-expressing strain in the presence of rapalog which should mislocate RIFIN away and therefore abolish the observed effect. If this does not happen, another expressed A RIFIN would exert the observed effect (which is possible anyway since rif transcription/expression is not monoallelic).”

“As a general question, why wasn’t the functionality of the “knock sideways” cassette not tried (mislocating tagged proteins via the FKBP/FRB domains)?”

We agree with the reviewer that the proposed experiment could address this issue however, the 2xFKBP domain has been removed from the construct during the cloning process (this is indicated in the methods section) as we wanted to avoid potential issues with having an disproportionately large tag of GFP and 2xFKBP. In addition the observations that the knock sideways method works often times very poorly with transmembrane proteins made us not include this tag.

“The authors comment that parasites selected to transcribe/express one specific stevor gene ceased to do so during generations without selection. Can this be shown also by Western blot or fluorescence microscopy?”

We performed limiting dilution on the Pf3D7_1149900.3HA cell line after taking it off neomycin pressure and obtained several clones which were tested for expression of the fusion protein via western blot (supplementary figure 3b). Supplying the drug back to the culture led to a drop in parasitemia and subsequent recovery of protein expression as detected by western blot (supplementary figure 3c). The parental pool was maintained under drug pressure. We tested for all 40 annotated stevor genes by RT-PCR and confirmed that expression had switched (supplementary Figure 3d).

“The expression “multigenes” in the discussion section is very unlucky and should be changed”

We have changed the wording in the discussion to highlight that we are targeting specific proteins within the multigene families and not the families as a whole.

“This observation however indicates that the SLI based knock-in of members of multigene families does not alter the expression dynamics but retains the control of the endogenous promoter.”

“In general, please revise the whole text for typographic mistakes starting in the title changing to “distinct members”. In a couple of places “Plasmodium” is written in small letters. In the introduction you mention small VSAs when you indeed mean the pir protein family, please do so. The last sentence in the introduction is too long and should be reformulated.”

“PLASMED in the Results section is also puzzling for the normal reader as is SCELSE (apparently a facility).”

We adjusted the text to clarify that PLASMED is an export signal motif and that SCELSE is a facility and revised the text for typographic errors.

“PLASMED is an export motif in rodent malaria parasites that has been shown to mediate protein export [69]“

“Triplicates of each RNA sample were outsourced to the genome facility at the Singapore Centre for Environmental Life Sciences Engineering (SCELSE) for RNA sequencing.”

“Please reformulate the Methods section in a way that it looks less like a lab journal. As an example, the description of the reverse transcription looks very much like a lab journal recipe”

We adjusted the manuscript accordingly.

“In Figure 1, I find the classification “host” misleading and would rather prefer “IRBC membranes” or “IRBC ghosts”. Also, the reader understands the message of Figure 1 easier if on the right side of the IFA pics the examined protein species is named, such “PfEMP1”, “RIFIN” and “STEVOR”. Please use also the same letter type in the figures and not mixtures of Arial, Times and Calibri.”

We agree that the “host” as label could be misleading and have changed it to exported proteins as the suggested labelling of Ghost or membranes is not quite correct since the fraction includes the soluble part of the host RBC as well. We agree with the formatting suggestions and changed them accordingly

“Although I believe that the primers used for qPCR were designed appropriately, the authors should invest a little more time and effort in describing how they were designed, and which sequence these had or alternatively a reference may be given.”

We added the sequences of the designed primers to the supplementary section (supplementary table 2-4)

“Regarding the supplementary material, please provide a plasmid map with oligo localization so that the reader can monitor what proves integration and what hints to the presence of residual episomal copies in the transfectants.”

We added a figure describing the SLI method which include the localization of the primers (Figure 1). A table with the integration validation primers has been added (supplementary table 3).

Reviewer 2

“1. A cartoon outlining the transfection strategy would help the reader navigate the manuscript.”

A cartoon explaining the SLI method and how it was utilized was added to the manuscript (Figure 1).

“2. What are the evidences that tagging the proteins does not affect their subcellular localization? Is the tagged PfEMP1 presented on the surface of the parasitized erythrocyte?”

It has been shown before that tagging with GFP still enables surface expression of PfEMP1 proteins (Melcher et al. 2010, Cellular Microbiology). Independent of that, it is to our knowledge not known whether every single var gene is exposed on the surface, especially considering that only a small subset of them has been studied yet. However, as already mentioned in response to reviewer 1, we performed the surface trypsinization experiment using a slightly adapted version of the protocol described in Baruch *et al.*, 1996, PNAS 93 which clearly shows that some of the tagged PfEMP1 is exposed on the surface of the iRBC (supplementary Figure 3a).

“3. It is unclear whether the activation of genes in close proximity to the integration site is a natural event, such that is it part of a biological mechanism, or whether it is an artefact resulting from the integration event and the forced gene activation. The activation of a second var gene in close proximity to the tagged gene would argue in favour of the latter hypothesis. A control might be to tag an unrelated gene adjacent to a var gene and see if this var gene is also activated.”

We agree that the integration event could lead to forced gene activation of adjacent genes however the observation by Golnitz et al., (Malaria Journal 7:14, 2008) shows that selection of specific var by panning on various CHO cell lines causes high expression of var Pf3D7_0420900 and Pf3D7_0421100 as well as the neighbouring var Pf3D7_0421300 transcript suggesting that a biological mechanism may be involved. Furthermore, we also show that activation of a var does not impact on the transcriptional status of a neighbouring rif providing some evidence that the integration event alone is not sufficient to lead to activation.

“4. The data presented suggest a cross-talk between different multi-gene families. Can the authors speculate about possible mechanisms?”

We added potential mechanisms of cross talk to the discussion.

“The importance of nuclear organization and non coding RNA on var gene expression has been already described in *Plasmodium falciparum* [81, 82]. Considering the importance of enhancer RNA on transcription in higher eukaryotes, we speculate that trans acting enhancers might play a more important role than discovered so far (reviewed in [83]). It is possible that similar mechanisms exists in *Plasmodium* which have not been discovered yet. Furthermore, the importance of non-coding RNA, whether from var introns or GC-rich ncRNAs has already been demonstrated in *P. falciparum* [74, 82]. It could be imagined that there are more, so far uncharacterized RNA species that a play a role in the interplay between multigene families.”

Reviewer 3

“Although the plasmids and the strategy have been described previously, some introduction on the original SLI method will help readers who do not know the method”

A cartoon explaining the SLI method and how it was utilized was added to the manuscript (Figure 1)

“Additionally, on page 4, second paragraph, the authors stated that “we have developed...” leading to thinking of a new approach developed by this work.”

We changed the wording to better indicate that we only utilize the SLI method in a novel way.

“In this study, we have adapted an experimental approach to overcome some of these critical challenges, thereby opening the potential to initiate a comprehensive and systematic study of VSAs.”

“Again, page 4, a plasmid map should be cited/provided when introducing the “resistant cassette”.”

A cartoon explaining the SLI method and how it was utilized was added to the manuscript (Figure 1).

“Some minor points:

1, Page 5, line 3, PFA: spell out.

2, Page 6, figure 1: Also label each panel of the images with STEVOR, pfEMP1, and RIFIN, respectively.

P3, age 7, second para, line 3: remove ‘,’ after (Pf3D7_1149900).

4, Figure 2a: It would help if the parasite groups (top) are also labeled.

5, Page 10, line 11: “var or stevor”.

6, Page 11, second paragraph, line 4: Replace “was analysed” with “were analysed”.

7, Page 11, second paragraph, line 9, “20 estimated counts”: Please explain. Line 2 from the bottom: Figure 4a should be Figure 5a. Expression data from only 60 yir genes are presented. What are the expression levels for other genes in the genome?

8, Figure 5b: Are the data adjusted with expression levels from YM?

9, Page 14, line 1: iRPMI? Spell out.

10, Page 14, last paragraph, Trophozoites staged parasites.

11, Page 17, line 5: PBS and 0.1% Tween-20.

12, Page 17, third paragraph, NFW filter: Please provide company name.

13, Page 19, line 3, Omelianzyk was supported by..?

14, Page 22, Supp. Table 1, TPM: spell out.”

Changes were made to the manuscript.

Reviewer 4

“1. Figure S1: In this integration PCR screen, a wild type band is still evident for all of the transgenic parasite lines. The authors interpret this band as irrelevant because it may represent other VSA genes, which may be amplified due to the high sequence homology around rifins, stevors, and var genes. However they haven’t shown that this is really the case. In my experience (and to my frustration), wild type DNA often persist after SLI selection with G418, even for single copy genes. Thus, cloning by limiting dilution of the parasite lines is still required in most cases. Therefore, the authors should use a different strategy to convincingly show integration of the transgene in the majority of the population, for example by FACS analysis, Southern blot or immunofluorescence analysis of the population (what proportion of parasites express the tagged variant?).”

We performed cell counts on IFAs of 1149900.3HA (stevor) and 0421300.GFP (var). For each cell line, 100+ parasites (at least 18hpi) were counted by three different members of the lab. Both parasite lines showed almost 100% staining for the tagged STEVOR and PfEMP1 (97 and 99.7% respectively) supporting integration of the transgene in this parasite populations (supplementary Figure 1d).

“2. Figure 1 Western Blot: I am confused by the statement that RIFINs and STEVORs are expected both in the pellet and SN fraction after saponin treatment (Results, section 1, paragraph 2). According to many publications, saponin fractionation under the conditions used by the authors leads to the separation of (infected) erythrocyte cytosol and soluble parasitophorous vacuole content from a pellet, which contains the parasite wrapped in the permeabilized host cell membranes. So RIFINs and STEVORs, which are transmembrane proteins throughout their export pathway, would be expected in the pellet. This is indeed consistent with the data shown in Figure 1. As a control the authors should run a WB for example on EXP2 or sEMP1 or some other Maurer’s cleft associated protein e.g. SBP1. Importantly, their conclusions about export of STEVOR and PfEMP1, and of parasite-restricted expression of RIFIN is still validated by the IFA images.”

We use those conditions for saponin extraction routinely in our lab. RBC transmembrane proteins can be detected in the supernatant under these conditions. In the Methods in Malaria Research 6th edition, page 97, chapter “Subcellular fractionation of iRBC: use of saponin and streptolysin O”, suggests the use of 0.1% saponin to “disintegration of the erythrocyte membrane and the parasitophorous vacuolar membrane”. We use slightly milder conditions to ensure that the parasite membrane remains intact. Furthermore, we perform slow speed centrifugation of only 900g to separate the parasite from the exported proteins. This is not fast enough to pellet the membranous compartments, including the ghost, from the saponin supernatant.

“3. The authors claim that transgenic stevor expression results in downregulation of all other stevor- but in fact there are several stevor variants that are upregulated in comparison to the 3D7 control according to Figure 2A (e.g. Pf3D7_0201300). Please comment on this. In my understanding this argues against mutually exclusive expression. However, this might also be a result of incomplete integration of the transgene at the stevor locus across the parasite population.”

The data obtained from microarray analysis is always a relative expression to a pool control. If a given gene is not expressed in the control, even trace amounts of RNA will give the impression of strong upregulation in the tested cell line. We performed qRT PCR on the cell line, testing for all 40 stevor genes annotated, and could confirm mutual exclusive expression (supplementary Figure 3d).

“4. Figure 3: According to the data shown in figure 2, the GFP tagged RIFIN variant is not upregulated relative to the 3D7 wild type control. Still, reduction of NK cell killing is only observed in the RIFIN transgenic parasite line, in which, however, the GFP tagged variant is not exported from the parasite. How do the authors explain these observations? “

We agree with this reviewer that this is indeed a surprising observation. However, we have previously shown the microvesicles exported by the parasite are able to modulate NK cell activity (Ye et al., 2018). While at this stage not able to provide direct proof that this is indeed the case we feel that microvesicle export of RIFIN is an attractive hypothesis to explain this observation. We have added this as a possible mechanism in the discussion.

“Surprisingly, the RIFIN does not appear to be exported beyond the parasite boundaries, excluding the possibility that the protein interacts with immune cells on the surface of the red blood cell. However, RIFIN have been recently detected within extracellular vesicles from field isolates and it has been shown that *P. falciparum* derived microvesicles can modulate NK-cell activity [76, 77]. It is therefore conceivable that the RIFIN is transported to its target cells through parasite derived extracellular vesicles.”

“5. Figure S2: Although there is clear evidence for integration, controls are missing that show that the wild type locus has been deleted from the parasite population and that equal amounts of DNA are analyzed (by PCR of an unmodified locus). This is critical for the interpretation of the gene expression results, as any parasites which are not subject to forced yir expression may express different dominant VSA variants.”

We have provided the controls in supplementary figure 4.

“6. Please include in the discussion other publications addressing the impact of sVSA promoter activation on the regulation of other VSA family members, e.g. Witmer et al 2012, Howitt et al 2009.”

We included the mentioned references to the discussion.

“Previous studies attempted to address this question by episomally expressing *stevor* introns and evaluating the effect on overall multigene family expression pattern with differing results [38, 72]. One obvious shortcoming of these approaches is that they will fail to detect regulatory mechanisms arising from genomic loci not included in the constructs such as enhancer like elements, GC-rich ncRNA or long non-coding RNA arising from introns, all of which can be found in *P. falciparum* [73-75].”

“Minor issues:

1. Introduction, line 1: The malaria parasite exports....

2. Figure 2a: please clarify what kind of data are shown in the heat map $-(\log)$ fold changes or expression levels?

3. Figure 2a: throughout the paper, the order of transgenic parasite lines is *Stevor – var- rifin*, but in this Figure the order is *var – stevor – rifin*. For consistency and clarity it would be helpful to change the order in the heatmap.

4. Figure 2d: It is unclear in this Figure which of the VSAs analyzed by qPCR belong to which family, please label them.

5. Figure 2d: Which tRNA synthetase was used for normalization?

6. Page 8: *Rifin Pf3D7_0421200* is oriented tail to tail, not head to head, with *var Pf3D7_0421300*.

***Rifin Pf3D7_0421500* is not directly adjacent to *var Pf3D7_0421300* but there is a *RUF6 ncRNA* gene**

(Pf3D7_0421200) in between.

7. Page 10 line 11: correct var or stevor

8. Page 10 2nd paragraph: what is PLASMED - should this be plasmid? Otherwise please explain.

9. Materials and Methods: NK cell killing assay – what media was used? “x 100” is missing in the formula

10. Primer sequences used for cloning/ qPCR/ diagnostic PCR should be provided as a supplement.”

We have made the relevant changes to the manuscript

Reviewers' comments:

Reviewer #1 (Remarks to the Author):

Most of the questions that I had were answered and a couple of new experiments were performed which in my opinion strengthen the manuscript.

Although I agree that the selection-linked method seems suitable for variant gene tagging, I still believe that it would have been more convincing to tag two var loci, one with a testable adhesive phenotype such as var2csa or var_{severe} and one with a yet unknown phenotype/a var gene never expressed in any report, such as the one which the authors tested.

Also, I still wonder if the activation of two head-to-tail var loci is an artifact. For example, in the paper of Dzikowki et al. (PLoS Pathog. 2006 Mar; 2(3): e22., Figure 3) the here "co-activated" var gene was detected "on" while the adjacent var (tagged and SLI-activated in the manuscript) was "off" in a cloned NF54 strain. Also in the Goelnitz paper (cited), when selecting on CHO-Selectin, the authors found few co-activation of Pf3D7_041100, while Pf3D7_041300 was strongly expressed. Regarding this, I would delete the suggestion about the opening of chromatin from the results section (page 7, last paragraph) and include in the discussion section a more critical view including that the described method may lead to artifacts such as diallelic expression (on page 14, last paragraph).

Again, and unfortunately, there is no description of a second targeted var locus (even if this was attempted). It remains to be seen if SLI in a subtelomeric upsA/upsB var locus may have led to different results. Since subtelomeric var loci are commonly attached to the nuclear periphery and localized in a silenced chromatin context, I wonder if integration at these sites is possible when using the SLI method. I seriously doubt that any var or rif gene can be tagged and selection-activated, and given the high similarity between ATS regions in var genes I am also quite sure that there is a great possibility of off-site tagging of a non-desired gene locus. This should somehow be considered in the discussion.

Minor points:

A couple of spelling errors persist which should still be corrected (I'm referring to the revised_marked_up manuscript file in pdf format).

Page 4 first line, change P. yoelli to P. yoelii.

In Figure 1, I would change the attribute "Recombination arm" for "homology region".

On page 5, I would change "Aldolase, a parasite internal protein, served as a..." to Aldolase, a cytosolic parasite protein, served as a..."

In Figure 2, the last sentence is truncated, possibly a formatting issue when using boxes around legends.

On page 8, in the middle of the last paragraph, include that "tRNA synthetase" was "seryl-tRNA synthetase". In the discussion section, the authors state that "Removal of the drug led to a slow decrease of gene expression up to the point where detection by Western blot was not possible anymore (data not shown)." Can the authors include the number of reinvasion cycles for this to happen?

In the last paragraph of the discussion, delete "To date" from the first sentence. In my opinion, and based on the tagging of only one member of P. falciparum variant gene families, the last paragraph is probably a little too optimistic.

In the methods section, on page 16, it must be "Transfection of Rodent malaria parasites".

Change also "...by i.p. injection when the parasitemia reaches 15%-60%" to past tense.

Regarding the authors' response to reviewer #4:

Reviewer 4

"1. Figure S1: In this integration PCR screen, a wild type band is still evident for all of the transgenic parasite lines. The authors interpret this band as irrelevant because it may represent other VSA genes, which may be amplified due to the high sequence homology around rifins, stevors, and var genes. However they haven't shown that this is really the case. In my experience (and to my frustration), wild type DNA often persist after SLI selection with G418, even for single copy genes. Thus, cloning by limiting dilution of the parasite lines is still required in most cases. Therefore, the authors should use a different strategy to convincingly show integration of the transgene in the majority of the population, for example by FACS analysis, Southern blot or immunofluorescence analysis of the population (what proportion of parasites express the tagged variant?)."

We performed cell counts on IFAs of 1149900.3HA (stevor) and 0421300.GFP (var). For each cell line, 100+ parasites (at least 18hpi) were counted by three different members of the lab. Both parasite lines showed almost 100% staining for the tagged STEVOR and PfEMP1 (97 and 99.7% respectively) supporting integration of the transgene in this parasite populations (supplementary Figure 1d).

R1: I had overlooked this detail in my analysis. I would have liked to see a Southern blot analysis such as was done for *P. yoelii* integrants. This would contain restricted DNA of the integrated version versus the episomal version, since the PCR fragment for unmodified loci is really very strong. A SB would be the gold standard to show integration. The authors may also have chosen another pair of oligos to show clean integration, namely one oligo anchoring in the plasmid amplifying the downstream integration site. In our hands, there is frequently an integration of a concatemeric sequence of the plasmid. FACS analysis is already an argument for successful integration but does not really prove what happened at the recombined locus.

"2. Figure 1 Western Blot: I am confused by the statement that RIFINs and STEVORs are expected both in the pellet and SN fraction after saponin treatment (Results, section 1, paragraph 2). According to many publications, saponin fractionation under the conditions used by the authors leads to the separation of (infected) erythrocyte cytosol and soluble parasitophorous vacuole content from a pellet, which contains the parasite wrapped in the permeabilized host cell membranes. So RIFINs and STEVORs, which are transmembrane proteins throughout their export pathway, would be expected in the pellet. This is indeed consistent with the data shown in Figure 1. As a control the authors should run a WB for example on EXP2 or sEMP1 or some other Maurer's cleft associated protein e.g. SBP1. Importantly, their conclusions about export of STEVOR and PfEMP1, and of parasite-restricted expression of RIFIN is still validated by the IFA images."

We use those conditions for saponin extraction routinely in our lab. RBC transmembrane proteins can be detected in the supernatant under these conditions. In the *Methods in Malaria Research* 6th edition, page 97, chapter "Subcellular fractionation of iRBC: use of saponin and streptolysin O", suggests the use of 0.1% saponin to "disintegration of the erythrocyte membrane and the parasitophorous vacuolar membrane". We use slightly milder conditions to ensure that the parasite membrane remains intact. Furthermore, we perform slow speed centrifugation of only 900g to separate the parasite from the exported proteins. This is not fast enough to pellet the membranous compartments, including the ghost, from the saponin supernatant.

R1: The answer is adequate and the methods used are sound.

"3. The authors claim that transgenic stevor3 expression results in downregulation of all other stevor- but in fact there are several stevor variants that are upregulated in comparison to the 3D7 control according to Figure 2A (e.g. Pf3D7_0201300). Please comment on this. In my

understanding this argues against mutually exclusive expression. However, this might also be a result of incomplete integration of the transgene at the stevor locus across the parasite population.”

The data obtained from microarray analysis is always a relative expression to a pool control. If a given gene is not expressed in the control, even trace amounts of RNA will give the impression of strong upregulation in the tested cell line. We performed qRT PCR on the cell line, testing for all 40 stevor genes annotated, and could confirm mutual exclusive expression (supplementary Figure 3d).

R1: There is no mentioning of Supplementary Figure 3 d in the revised text. Also, the legend of this figure is confusing and the authors should specify how the clones analyzed here were treated. It is also puzzling that the relative stevor transcript quantity of clone G2 is so high compared to C5.

“4. Figure 3: According to the data shown in figure 2, the GFP tagged RIFIN variant is not upregulated relative to the 3D7 wild type control. Still, reduction of NK cell killing is only observed in the RIFIN transgenic parasite line, in which, however, the GFP tagged variant is not exported from the parasite. How do the authors explain these observations? ”

We agree with this reviewer that this is indeed a surprising observation. However, we have previously shown the microvesicles exported by the parasite are able to modulate NK cell activity (Ye et al., 2018). While are at this stage not able to provide direct proof that this is indeed the case, we feel that microvesicle export of RIFIN is an attractive hypothesis to explain this observation. We have added this as a possible mechanism in the discussion.

“Surprisingly, the RIFIN does not appear to be exported beyond the parasite boundaries, excluding the possibility that the protein interacts with immune cells on the surface of the red blood cell. However, RIFIN have been recently detected within extracellular vesicles from field isolates and it has been shown that *P. falciparum* derived microvesicles can modulate NK-cell activity [76, 77]. It is therefore conceivable that the RIFIN is transported to its target cells through parasite derived extracellular vesicles.”

R1: This is an acceptable answer.

“5. Figure S2: Although there is clear evidence for integration, controls are missing that show that the wild type locus has been deleted from the parasite population and that equal amounts of DNA are analyzed (by PCR of an unmodified locus). This is critical for the interpretation of the gene expression results, as any parasites which are not subject to forced yir expression may express different dominant VSA variants.”

We have provided the controls in supplementary figure 4.

R1: This is sufficient, although a control showing the digested transfection plasmid would be desirable.

“6. Please include in the discussion other publications addressing the impact of sVSA promoter activation on the regulation of other VSA family members, e.g. Witmer et al 2012, Howitt et al 2009.”

We included the mentioned references to the discussion.

R1: OK.

Reviewer #2 (Remarks to the Author):

The authors have fully addressed my concerns. I have no more suggestions.

Reviewer #1 (Remarks to the Author):

Most of the questions that I had were answered and a couple of new experiments were performed which in my opinion strengthen the manuscript.

Although I agree that the selection-linked method seems suitable for variant gene tagging, I still believe that it would have been more convincing to tag two var loci, one with a testable adhesive phenotype such as var2csa or varsevere and one with a yet unknown phenotype/a var gene never expressed in any report, such as the one which the authors tested.

Reply:

Activation of a functional PfEMP1 would extend the focus of the manuscript further. However, PfEMP1 are not the main focus of this study. The var gene was chosen to ensure that expression is driven by the SLI construct and not by phenotypic selection. And there is a potential complication in relation to var2csa, which has been proposed to be a “steady-state” var (Kyes et al 2003) and therefore could lead to misleading results. Importantly, functional expression of multigene family members was clearly shown for the RIFIN.

Also, I still wonder if the activation of two head-to-tail var loci is an artifact. For example, in the paper of Dzikowki et al. (PLoS Pathog. 2006 Mar; 2(3): e22., Figure 3) the here "co-activated" var gene was detected "on" while the adjacent var (tagged and SLI-activated in the manuscript) was "off" in a cloned NF54 strain. Also in the Goelnitz paper (cited), when selecting on CHO-Selectin, the authors found few co-activation of Pf3D7_041100, while Pf3D7_041300 was strongly expressed. Regarding this, I would delete the suggestion about the opening of chromatin from the results section (page 7, last paragraph) and include in the discussion section a more critical view including that the described method may lead to artifacts such as diallelic expression (on page 14, last paragraph).

Reply:

We agree that the observed phenotype is different from what has been observed in relation to published work and we have therefore adjusted the text along the recommendations made by this reviewer.

“It could be speculated that the open chromatin necessary for expression of a multigene member stretches for several kilobases both up- and downstream and can lead to the activation of adjacent genes. At this point in time though it cannot be ruled out that this is a result of the introduction of the plasmid into the genomic locus. However, introduction of hDHFR into a subtelomeric region of chromosome 3 did not lead to activation of adjacent var [29]. This would suggest that introduction of a drug selectable marker into the vicinity of a var alone does not cause expression of that gene.”

Again, and unfortunately, there is no description of a second targeted var locus (even if this was attempted). It remains to be seen if SLI in a subtelomeric upsA/upsB var locus may have led to different results. Since subtelomeric var loci are commonly attached to the nuclear periphery and localized in a silenced chromatin context, I wonder if integration at these sites is possible when using the SLI method. I seriously doubt that any var or rif gene can be tagged and selection-activated, and given the high similarity between ATS regions in var genes I am also quite sure that there is a great

possibility of off-site tagging of a non-desired gene locus. This should somehow be considered in the discussion.

As mentioned in the response above the focus of this study is not in relation to *var* gene activation that has been extensively explored in the field. It is rather to focus on multigene families where to date it was not possible to select for the expression of a unique member in a parasite population. In relation to the activation of genes located in silenced subtelomeric regions our data would indicate that this is possible as both the *rif* and *stevor* activated using the SLI method are found in these regions.

Minor points:

A couple of spelling errors persist which should still be corrected (I'm referring to the revised_marked_up manuscript file in pdf format).

Page 4 first line, change P. yoelli to P. yoelii.

In Figure 1, I would change the attribute "Recombination arm" for "homology region".

On page 5, I would change "Aldolase, a parasite internal protein, served as a..." to Aldolase, a cytosolic parasite protein, served as a..."

In Figure 2, the last sentence is truncated, possibly a formatting issue when using boxes around legends.

On page 8, in the middle of the last paragraph, include that "tRNA synthetase" was "seryl-tRNA synthetase". In the discussion section, the authors state that "Removal of the drug led to a slow decrease of gene expression up to the point where detection by Western blot was not possible anymore (data not shown)." Can the authors include the number of reinvasion cycles for this to happen?

In the last paragraph of the discussion, delete "To date" from the first sentence. In my opinion, and based on the tagging of only one member of P. falciparum variant gene families, the last paragraph is probably a little too optimistic.

In the methods section, on page 16, it must be "Transfection of Rodent malaria parasites".

Change also "...by i.p. injection when the parasitemia reaches 15%-60%." to past tense.

Reply:

We thank the reviewer for their careful reading. We have made the changes in the manuscript accordingly.

We highlighted the need for extensive controls in last paragraph of the discussion:

"Despite the need for thorough controls, the ability to activate expression of so far understudied genes will significantly enhance our ability to study the function as well as regulation of different multigene families in Plasmodium."

Regarding the authors' response to reviewer #4:

Reviewer 4

"1. Figure S1: In this integration PCR screen, a wild type band is still evident for all of the transgenic parasite lines. The authors interpret this band as irrelevant because it may represent other VSA genes, which may be amplified due to the high sequence homology around rifins, stevors, and var genes. However they haven't shown that this is really the case. In my experience (and to my frustration), wild type DNA often persist after SLI selection with G418, even for single copy genes. Thus, cloning by limiting dilution of the parasite lines is still required in most cases. Therefore, the authors should use a different strategy to convincingly show integration of the transgene in the majority of the population, for example by FACS analysis, Southern blot or immunofluorescence analysis of the population (what proportion of parasites express the tagged variant?)."

We performed cell counts on IFAs of 1149900.3HA (stevor) and 0421300.GFP (var). For each cell line, 100+ parasites (at least 18hpi) were counted by three different members of the lab. Both parasite lines showed almost 100% staining for the tagged STEVOR and PfEMP1 (97 and 99.7% respectively) supporting integration of the transgene in this parasite populations (supplementary Figure 1d).

R1: I had overlooked this detail in my analysis. I would have liked to see a Southern blot analysis such as was done for P. yoelii integrants. This would contain restricted DNA of the integrated version versus the episomal version, since the PCR fragment for unmodified loci is really very strong. A SB would be the gold standard to show integration. The authors may also have chosen another pair of oligos to show clean integration, namely one oligo anchoring in the plasmid amplifying the downstream integration site. In our hands, there is frequently an integration of a concatemeric sequence of the plasmid. FACS analysis is already an argument for successful integration but does not really prove what happened at the recombined locus.

Reply:

In case of the stevor Pf3D7_1149900 we were able to find an oligo pair which shows loss of the unmodified locus after integration. The manuscript has been modified accordingly, changing supplementary figure S1b and supplementary table S3 to reflect the new pair of oligos.

As we were unable to find clean oligos for the var Pf3D7_0421300 and rifin Pf3D7_1254800, we decided to perform single cell cloning on those two cell lines. We sent 2 clones of each of the three cell lines (the var, rifin and stevor knock-in cell lines) for whole genome sequencing using the Oxford Nanopore GridION platform (i.e. ONT). Genomic DNA from the six clones was purified by phenol chloroform extraction, followed by ethanol precipitation. The purified DNA was passed to the Genome Institute Singapore (GIS). GIS finished the library preparation by adapter ligation and performed multiplexed sequencing of all libraries on a single flowcell. Reads were demultiplexed and analysed as independent libraries.

The 3D7 genome was downloaded from PlasmO_DB and for each construct, the entire plasmid sequence was inserted at the intended locus, resulting in three different edited genomes. Demultiplexed ONT reads were mapped to the appropriate edited genome using minimap2 and sorted using samtools. Mapped reads were visualized using the Integrated Genomics Viewer (Version 2.8.2).

ONT reads mapping to the targeted locus covered either only the plasmid (due to episomal plasmid remaining within the cell lines), covered the targeted locus and wild-type 3' UTR, but not the plasmid sequence (i.e. WT remaining in the cell) or covered the 5' and 3' beyond the boundaries of the plasmid. Every single read which extends beyond the boundaries of the plasmid integrated into the correct locus, confirming integration into the targeted gene.

In order to assess integration of the plasmid into another locus, we extracted all ONT reads covering the plasmid sequence. Since default mapping with minimap2 allows for soft-clipping, the end of those reads potentially map to other genomic regions and reveal off-target integration.

Thus, the 200bps at both ends of every ONT read covering the plasmid were extracted. These short reads were mapped using bwa again against the entire genome and the genic loci that these fragments aligned to were quantified. The majority of reads map to the correct, targeted locus. Only a small subset of read ends align to other parts of genome (summarized as “others”). Due to the short length of those read-ends and the high error rate of nanopore sequencing, those reads represent supplementary mappings that map equally well to the correct, intended locus and to other regions within the genome. In addition, while these off target often have only a single read mapping to them, the targeted locus is covered on average by 244 reads.

In summary, we performed whole genome sequencing on two clones from each *P. falciparum* knock-in cell line using Nanopore sequencing. Every read extending beyond the plasmid itself maps to the correct locus, showing correct integration. Mapping of the ends of reads spanning the plasmid confirmed this finding. We therefore conclude that the plasmid only integrated into the correct locus in the genome.

We added a description of the whole genome sequencing to the materials and methods section.

“Whole genome sequencing and analysis

Genomic DNA of two clones of all three *P. falciparum* cell lines was extracted using phenol:chloroform:isoamylalcohol (25:24:1, Invitrogen) followed by ethanol precipitation as per manufacturers protocol. Genomic DNA was passed to the Genome Institute of Singapore for library preparation and sequencing. DNA libraries were prepared by 1D native, PCR free barcode ligation. Multiplexed libraries were sequenced on an Oxford Nanopore GridION platform release 19.12.6 using a single flowcell (Basecalling/mode: Guppy 3.2.10/High accuracy). *Plasmodium falciparum* 3D7 reference genomes were extracted from PlasmoDB and edited to introduce the appropriate plasmid sequence at the gene of reference, resulting in three different edited genomes. Reads were demultiplexed and mapped to the according edited reference genome using minimap2 and sorted with Samtools. Mapped reads were visualized using IGV (Version 2.8.2). In order to quantify correct integration events, 200bps of both ends from every read containing the plasmid were mapped against the reference genome using a cut-off of $1e^{-20}$. Mapped loci were counted and percentage of correct integration events calculated.”

We added the quantification of read-end mapping to supplementary figure 1e.

“3. The authors claim that transgenic stevor expression results in downregulation of all other stevor- but in fact there are several stevor variants that are upregulated in comparison to the 3D7 control according to Figure 2A (e.g. Pf3D7_0201300). Please comment on this. In my understanding this argues against mutually exclusive expression. However, this might also be a result of incomplete integration of the transgene at the stevor locus across the parasite population.”

The data obtained from microarray analysis is always a relative expression to a pool control. If a given gene is not expressed in the control, even trace amounts of RNA will give the impression of strong upregulation in the tested cell line. We performed qRT PCR on the cell line, testing for all 40 stevor genes annotated, and could confirm mutual exclusive expression (supplementary Figure 3d).

R1: There is no mentioning of Supplementary Figure 3 d in the revised text. Also, the legend of this figure is confusing and the authors should specify how the clones analyzed here were treated. It is also puzzling that the relative stevor transcript quantity of clone G2 is so high compared to C5.

Reply:

We have added a comment to the results section mentioning supplementary figures 3b-d.

“Single cell cloning of the stevor knock-in cell line in absence of neomycin identified several clones which stopped expression of the fusion protein (supplementary figure 3b, d). Addition of the drug forced the parasite to resume expression of Pf3D7_1149900 (supplementary figure 3c).”

Changed the legend of supplementary figure for clarity.

“Supp. figure 3: Further characterization of knock-in cell lines. (a) Pf3D7_0421300 is partially presented on the surface of the infected red blood cell. Percoll enriched late stage parasites expressing Pf3D7_0421300.GFP were subjected to live trypsin treatment. After treatment, an additional band corresponding to the size of PfEMP1 ATS and GFP is detectable on a western blot (green arrow). (b-d) Knock-in parasites stop expression of the targeted protein without drug pressure. Neomycin was removed from the culture medium of 1149900.3HA parasites and single clones were obtained by limiting dilution. (b) The clones obtained from limiting dilution were tested for expression of the fusion protein by western blot using Histone 3 as loading control. Four of the clones (C5, C6, G2, G6) showed no expression of the HA-tagged STEVOR. Two of the clones (C5, G2, highlighted with black arrows) were chosen for further studies. (c) Expression of the fusion protein can be recovered upon addition of the drug. Neomycin was added back to the culture medium of the non-expressing clones C5 and G2. After an initial drop in parasitaemia, late stage parasites were collected and analysed using western blot. Expression of the fusion protein could be detected in both clones. (d) Clonal dilution identified parasites expressing different stevor. The two chosen clones were tested for their expression of all 40 annotated stevor genes and the previously tested var and rif (labelled with v or r respectively). In accordance to the western blot results, neither of the clones shows detectable levels of Pf3D7_1149900 expression. Furthermore, expression of the rif Pf3D7_1149800 dropped below detection limits as well. Expression of the var Pf3D7_0617400 further increased compared to the unselected pool. Expression of only the targeted gene can be detected in the pool (black arrow). Clone C5 expresses two stevor (Pf3D7_0832000 and Pf3D7_0617600) while clone G2 expresses only Pf3D7_1040200 at very high levels. Considering that stevor expression in vitro is generally reduced, the high expression levels observed in clone G2 might more closely reflect in vivo stevor expression.”

REVIEWERS' COMMENTS:

Reviewer #1 (Remarks to the Author):

The manuscript by Omelianczyk was now improved and shows the application of selection linked integration in four different loci of variant gene families in two different Plasmodium species, Plasmodium falciparum and Plasmodium yoelii. The authors show the proof of principle and provide thorough evidence that integration in fact occurred, turning this technique useful for the intended purpose, the study of members of multigene families and other genes in these organisms. I still believe that for the important var family the inclusion of upsA and upsB var gene loci would have been ideal since members of their group are associated to significant virulence (e.g. var severe, var2csa, var3), and the modification of a single upsC locus is the minimal, though probably sufficient, effort to show the functionality of the system.

The methods used and their description are appropriate and permit reproduction. The data analysis (RNAseq) was done using the indicated applications.

Details:

Comments on the revised manuscript by Omelianczyk et al.

The revised version of the manuscript by Omelianczyk and colleagues has been revised and many of the points raised during review of the initial manuscript were improved.

I believe that the data were thoroughly analyzed and critically approached, however I am still sure that the tagging of any variant gene, specifically of the var and rif groups, is probably impossible. I would highly recommend introducing this point. If the expression of any variant gene is too late in the cycle or cannot surpass a determined transcript level for whatever reason (including "defect promoters" after a natural ectopic recombination event), integrated versions may never be obtained.

Also, in the case of var genes, the ATS region used for tagging is quite similar between different var genes and off-target integration in other (perhaps also interesting) var loci may result. I strongly recommend mentioning all this in the beginning of the discussion section.

In their reply letter, the authors replied to my suggestion to target as a proof of principle also a var gene with known phenotype when expressed, such as var2csa. In their response, however, they argue that this var gene is continuously expressed, which is not true. They refer to var1csa, which is apparently transcribed but does not result in a phenotype. It is true that var2csa seems to be a locus activated by default and then downregulated at the translational level apparently by an untranslated ORF in the 5' upstream region of var2csa (work from Kirk Deitsch's group). I still find the targeting of upsA and B var genes attractive and wonder if SLI targeting works in subtelomeric var genes.

I liked the thorough way, how the authors show now that integration in fact occurred at the desired loci. This is obviously much better than the previous PCR pictures.

Some minor points should be changed such as in the first paragraph of the introduction regarding the statement about PfEMP1, which are not always larger than 250 kDa, in fact, there are smaller ones.

Please change to "high molecular weight...".

Then in the second paragraph, there is a typo, it must be Trypanosoma brucei.

In the discussion section, the authors write that several authors "episomally expressed stevor introns", obviously this must be "stevor genes" or "stevor exons".

Reviewer #1 (Remarks to the Author):

The manuscript by Omelianczyk was now improved and shows the application of selection linked integration in four different loci of variant gene families in two different Plasmodium species, Plasmodium falciparum and Plasmodium yoelii. The authors show the proof of principle and provide thorough evidence that integration in fact occurred, turning this technique useful for the intended purpose, the study of members of multigene families and other genes in these organisms. I still believe that for the important var family the inclusion of upsA and upsB var gene loci would have been ideal since members of their group are associated to significant virulence (e.g. var severe, var2csa, var3), and the modification of a single upsC locus is the minimal, though probably sufficient, effort to show the functionality of the system.

The methods used and their description are appropriate and permit reproduction. The data analysis (RNAseq) was done using the indicated applications.

Details:

Comments on the revised manuscript by Omelianczyk et al.

The revised version of the manuscript by Omelianczyk and colleagues has been revised and many of the points raised during review of the initial manuscript were improved.

I believe that the data were thoroughly analyzed and critically approached, however I am still sure that the tagging of any variant gene, specifically of the var and rif groups, is probably impossible. I would highly recommend introducing this point. If the expression of any variant gene is too late in the cycle or cannot surpass a determined transcript level for whatever reason (including “defect promoters” after a natural ectopic recombination event), integrated versions may never be obtained.

Also, in the case of var genes, the ATS region used for tagging is quite similar between different var genes and off-target integration in other (perhaps also interesting) var loci may result. I strongly recommend mentioning all this in the beginning of the discussion section.

Reply:

We added another passage to the discussion, addressing the issues mentioned and further cautioning for thorough controls.

“Considering the high homology of the C-terminal regions among members of multigene families, careful validation of correct integration is of essence. Generation of tagged cell lines relies on expression of the gene beyond a minimal threshold, potentially excluding genes with low switch rates or defective promoters.”

In their reply letter, the authors replied to my suggestion to target as a proof of principle also a var gene with known phenotype when expressed, such as var2csa. In their response, however, they argue that this var gene is continuously expressed, which is not true. They refer to var1csa, which is apparently transcribed but does not result in a phenotype. It is true that var2csa seems to be a locus activated by default and then downregulated at the translational level apparently by an untranslated ORF in the 5' upstream region of var2csa (work from Kirk Deitsch's group). I still find the targeting of upsA and B var genes attractive and wonder if SLI targeting works in subtelomeric var genes.

I liked the thorough way, how the authors show now that integration in fact occurred at the desired loci. This is obviously much better than the previous PCR pictures.

Some minor points should be changed such as in the first paragraph of the introduction regarding the statement about PfEMP1, which are not always larger than 250 kDa, in fact, there are smaller ones. Please change to "high molecular weight..."

*Then in the second paragraph, there is a typo, it must be *Trypanosoma brucei*.*

In the discussion section, the authors write that several authors "episomally expressed stevor introns", obviously this must be "stevor genes" or "stevor exons".

Reply:

The sentences have been changed accordingly

"VSAs can be broadly grouped into the high molecular weight and *P. falciparum* specific PfEMP1 (coded by *var* genes) [6] and the small variant surface antigens (sVSAs) which are usually about 30 – 40 kDa in size and found in all *Plasmodium spp* [5, 7-10]."

"Even though it has been shown that multigene families in other organisms such as the VSGs from *Trypanosoma brucei*, serotypes of *Borrelia hermsii* or mating types in *Saccharomyces cerevisiae* show high levels of regulation [17-19], only *var* genes seem to show the same level of mutual exclusive control."

"Previous studies attempted to address this question by episomally expressing *stevor* exons and evaluating the effect on overall multigene family expression pattern with differing results [38, 72]."